

# The landscape of unfolding with machine learning

Nathan Huetsch[1], Javier Mariño Villadamigo[1], Alexander Shmakov[2],
Sascha Diefenbacher[3], Vinicius Mikuni[3], Theo Heimel[1],
Michael Fenton[2], Kevin Greif[2], Benjamin Nachman[3,4],
Daniel Whiteson[2], Anja Butter[1,5], and Tilman Plehn[1,6]

**1** Institut für Theoretische Physik, Universität Heidelberg, Germany
**2** Department of Physics and Astronomy, University of California, Irvine, USA
**3** Physics Division, Lawrence Berkeley National Laboratory, Berkeley, USA
**4** Berkeley Institute for Data Science, University of California, Berkeley, USA
**5** LPNHE, Sorbonne Université, Université Paris Cité, CNRS/IN2P3, Paris, France
**6** Interdisciplinary Center for Scientific Computing (IWR), Universität Heidelberg, Germany

## Abstract

Recent innovations from machine learning allow for data unfolding, without binning and including correlations across many dimensions. We describe a set of known, upgraded, and new methods for ML-based unfolding. The performance of these approaches are evaluated on the same two datasets. We find that all techniques are capable of accurately reproducing the particle-level spectra across complex observables. Given that these approaches are conceptually diverse, they offer an exciting toolkit for a new class of measurements that can probe the Standard Model with an unprecedented level of detail and may enable sensitivity to new phenomena.

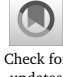

## 1 Introduction

Particle physics experiments seek to reveal clues about the fundamental properties of particles and their interactions. A key challenge is that predictions from quantum field theory are at the level of partons, while experiments observe the corresponding detector signatures. Precise and detailed simulations link these two levels [1]. They fold predictions for the hard process through QCD effects, hadronization, and the detector response to compare with data. This statistically powerful *forward inferences* approach has been widely used.

However, forward inference requires access to the data and accurate detector simulations. These conditions are rarely satisfied outside of a given experiment, severely limiting the ability of the broader community to study particle physics data. In addition, analysis of data from the high-luminosity LHC with forward inference will require precise simulations for every hypothesis, challenging available computing resources.

An alternative approach is *unfolding*. Rather than correcting predictions for the effects of the detector, the data are adjusted to provide an estimate of their pre-detector distributions. Since the effects described by our forward simulation are stochastic, this adjustment is performed on a statistical basis. Unfolding offers important advantages, such as making data analysis possible by a broader community and enabling an efficient combination of data from several experiments, such as in global analyses of the Standard Model Effective Theory [2, 3].

Traditional unfolding algorithms have been used extensively, successfully delivering a multitude of differential cross section measurements [4–7]. The most widely-used methods are Iterative Bayesian Unfolding [8–11], Singular Value Decomposition [12], and TUnfold [13]. However, each of these methods can only be applied to binned datasets of small dimensionality, such that the unfolded observables and their binning have to be selected in advance.

Machine learning (ML) techniques have revolutionized unfolding by allowing for unbinned cross sections to be measured across many dimensions [6, 14]. Where sufficient information is unfolded, new observables can be calculated from unbinned data, long after the initial publication. The first ML-based unfolding method applied to data is OmniFold [15, 16], which uses classifiers to reweight simulations. It has recently been applied to studies of hadronic final states at H1 [17–20], LHCb [21], CMS [22], and STAR [23]. Alternative ML-unfolding methods use generative networks, either for distribution mapping [24–27] or for probabilistic, conditional generation [28–35].

The goal of this paper is to lay out and extend the landscape of ML methods. We benchmark a diverse set of approaches on the same datasets, to facilitate direct comparisons. Some methods have been studied with an iterative component to mitigate the sensitivity to starting particle-level simulations. To simplify the setup and reduce stochastic effects from iterating, we apply all methods with only a single step. The goal is to estimate the posterior with the starting simulation as the prior. Performing this step well is the essential component of a full unfolding approach.

We begin with a brief introduction of the different methods for ML-based unfolding in Sec. 2. In Sec. 3, we show how all approaches can accurately unfold from detector level to the particle level using a $Z$+jets benchmark dataset. For certain theory questions it is useful to further unfold to the parton level, treating QCD radiation as a distortion to be corrected like detector effects. As an example of this type of unfolding, we study top quark pair production in Sec. 4. In Sec. 5 we summarize the advantages of the different methods, to help the experimental collaborations pick the method(s) best-suited for a given task. The figures shown in App. A combine results from the $Z$+jets study in Sec. 3.

## 2 ML-unfolding

We define our unfolding problem using four phase space densities, which are encoded in the corresponding samples, in the sense of unsupervised density estimation in ML-terms. We rely on simulated predictions at the particle/parton level, $p_{\text{gen}}(x_{\text{part}})$, and the detector or reconstruction (reco) level, $p_{\text{sim}}(x_{\text{reco}})$. Unfolding turns the measured $p_{\text{data}}$ into $p_{\text{unfold}}$,

$$
\begin{array}{ccc}
p_{\text{gen}} & \xleftrightarrow{\text{unfolding inference}} & p_{\text{unfold}}(x_{\text{part}}) \\
{\scriptstyle\text{simulation}}\downarrow & & \uparrow{\scriptstyle\text{unfolding}} \\
p_{\text{sim}} & \xleftarrow{\text{forward inference}} & p_{\text{data}}(x_{\text{reco}})
\end{array}
\tag{1}
$$

Our simulated samples come in pairs $(x_{\text{part}}, x_{\text{reco}})$, which can be used for unfolding. Data only exist on the $x_{\text{reco}}$ level. The features of the unfolded data $p_{\text{unfold}}$ should be determined by $p_{\text{data}}$, but will always include a data-independent bias from the assumed $p_{\text{gen}}$. The question how we can minimize the resulting model dependence will be part of a follow-up of this study.

Established ML-techniques for unfolding rely on one of two approaches. They either reweight simulated samples, or they generate unfolded samples from conditional probabilities. We will briefly introduce both original methods [15, 28, 29], as well as a more recent hybrid approach of mapping distributions using generative networks.

### 2.1 Reweighting: OmniFold and bOmnifold

The deep learning-based approach to unfolding via re-weighting is OmniFold [15, 16]. It is based on the Neyman–Pearson lemma [36], stating that an optimally trained, calibrated classifier $C$ will learn the likelihood ratio of the two underlying phase space distributions. If we use a binary cross entropy (BCE) loss to distinguish between data and simulated reco-level events, then the following combination approximates the likelihood ratio:

$$
w(x_{\text{reco}}) \equiv \frac{p_{\text{data}}(x_{\text{reco}})}{p_{\text{sim}}(x_{\text{reco}})} = \frac{C(x_{\text{reco}})}{1 - C(x_{\text{reco}})} \, .
\tag{2}
$$

OmniFold computes these classifier weights at the reco-level, and uses the paired simulated data to pull these weights from the reco-level events to the particle-level events. The re-

weighted simulated events then define

$$p_{\text{unfold}}(x_{\text{part}}) = w(x_{\text{reco}})\, p_{\text{gen}}(x_{\text{part}}).\tag{3}$$

This weight-pushing is the first step in the two-step iterative OmniFold algorithm. Because we are leaving out the model dependence to a dedicated second study, we restrict ourselves to this first iteration, which in the scheme of Eq.(1) looks like

$$
\begin{array}{ccc}
p_{\text{gen}} & \xrightarrow{\text{apply reweighting}} & p_{\text{unfold}}(x_{\text{part}}) \\
\text{pull weights}\Big\uparrow & & \\
p_{\text{sim}} & \xleftarrow{\text{train reweighting}} & p_{\text{data}}(x_{\text{reco}})
\end{array}
\tag{4}
$$

**Bayesian neural network (BNN)** Bayesian versions can be derived for any deterministic neural network with a likelihood loss [37–41]. The BNN training does not fix the network parameters, but allows them to learn distributions, such that sampling over the network parameters gives the probability distribution in model space, i.e. for the network output. Based on studies for regression [42,43] and classification tasks [44], there is evidence that for a sufficiently deep network we can assign independent Gaussians to each network parameter [39]. This effectively doubles the size of the network which now learns a central prediction and the error bar simultaneously. Even though the weights are Gaussian distributed, the final network output is generally not a Gaussian. As we will see below, Bayesian networks can be generalized to generative tasks [45–47].

One benefit of Bayesian networks is that they automatically include a generalized dropout and a weight regularization [41, 48, 49], derived from Bayes' theorem together with the likelihood loss. This means that BNNs are automatically protected from overtraining and an attractive option for applications where the precision of the network is critical, like the classifier reweighting in OmniFold.

## 2.2 Mapping distributions: Schrödinger bridge and direct diffusion

Instead of reweighting phase-space events, we can use generative neural networks to morph a base distribution to a target distribution. In our case, we train a network to map event distributions from $x_{\text{reco}}$ to $x_{\text{part}}$ based on the paired or unpaired simulated events and apply this mapping to $p_{\text{data}}(x_{\text{reco}})$ to generate $p_{\text{unfold}}(x_{\text{part}})$:

$$
\begin{array}{ccc}
p_{\text{gen}} & & p_{\text{unfold}}(x_{\text{part}}) \\
\text{training}\Big\uparrow & & \Big\uparrow \text{distribution mapping} \\
p_{\text{sim}} & \xleftarrow{\text{correspondence}} & p_{\text{data}}(x_{\text{reco}})
\end{array}
\tag{5}
$$

As mentioned above, the trained mapping assumes that $p_{\text{sim}}$ and $p_{\text{data}}$ describe the same features at the reco-level. Two ML-methods that we study for this task include Schrödinger Bridges [26] and Direct Diffusion [27], see also Ref. [24] for an early study.

### 2.2.1 Schrödinger bridge

Schrödinger Bridges define the transformation between particle-level events $x_{\text{part}} \sim p_{\text{gen}}$ to reco-level events $x_{\text{reco}} \sim p_{\text{sim}}$ as a time-dependent process following a forward-time stochastic

differential equation (SDE)

$$dx = f(x,t)dt + g(t)dw. \tag{6}$$

The drift term $f$ controls the deterministic part of the time-evolution, $g$ is the noise schedule, and $dw$ a noise infinitesimal. For such an SDE, the reverse time evolution follows the SDE

$$dx = [f(x,t) - g(t)^2 \nabla \log p(x,t)]dt + g(t)dw, \tag{7}$$

with the corresponding score $s(x,t) = \nabla \log p(x,t)$. To construct an unfolding, we need to find $f$ and $g$ for our forward process from particle level to reco level, and then encode the score function in the unfolding network $s_\theta(x,t)$ [50].

Constructing a forward-time SDE that transforms an arbitrary distribution into another is much more challenging than mapping a distribution into a noise distribution with known probability density (e.g. a Gaussian), as is the case for standard SDE-based diffusion networks. A framework to construct a transport plan in the general case was proposed by Erwin Schrödinger [51]. It introduces two wave functions describing the time-dependent density as $p(x,t) = \widehat{\Psi}(x,t)\Psi(x,t)$. By setting the drift coefficient to $f = g(t)^2 \nabla \log \Psi(x,t)$ the forward and reverse SDEs in Eqs.(6) and (7) become

$$\begin{aligned} dx &= \phantom{-}g(t)^2 \nabla \log \Psi(x,t)dt + g(t)dw, \\ dx &= -g(t)^2 \nabla \log \widehat{\Psi}(x,t)dt + g(t)dw. \end{aligned} \tag{8}$$

If the two wave-functions fulfill the coupled partial differential equations

$$\begin{aligned} \frac{\partial \Psi(x,t)}{\partial t} &= -\frac{1}{2}g(t)^2 \Delta \Psi(x,t), \\ \frac{\partial \widehat{\Psi}(x,t)}{\partial t} &= \phantom{-}\frac{1}{2}g(t)^2 \Delta \widehat{\Psi}(x,t), \end{aligned} \tag{9}$$

with the boundary conditions

$$\widehat{\Psi}(x,t)\Psi(x,t) = \begin{cases} p_{\text{gen}}(x), & t = 0, \\ p_{\text{sim}}(x), & t = 1, \end{cases} \tag{10}$$

then the SDEs in Eq.(8) transform particle-level events to reco-level events and vice versa.

Next, we need to find $\Psi$, $\widehat{\Psi}$ that fulfill the conditions. The authors of Ref. [52] observe that reverse generation following Eq.(8) does not require access to the wave functions, but only to the score function $\nabla \log \widehat{\Psi}$. For paired training data,

$$(x_0, x_1) \sim (p_{\text{gen}}, p_{\text{sim}}), \tag{11}$$

the posterior encoded in the SDEs in Eq.(8), conditioned on the respective initial and final states, has the analytic form

$$q(x|x_0, x_1) = \mathcal{N}(x_t; \mu_t(x_0, x_1), \Sigma_t),$$

$$\text{with} \quad \mu_t(x_0, x_1) = \frac{\bar{\sigma}_t^2}{\bar{\sigma}_t^2 + \sigma_t^2}x_0 + \frac{\sigma_t^2}{\bar{\sigma}_t^2 + \sigma_t^2}x_1, \quad \text{and} \quad \Sigma_t = \frac{\sigma_t^2 \bar{\sigma}_t^2}{\bar{\sigma}_t^2 + \sigma_t^2}, \tag{12}$$

denoting $\sigma_t^2 = \int_0^t g^2(\tau)d\tau$ and $\bar{\sigma}_t^2 = \int_t^1 g^2(\tau)d\tau$. This allows for the generation of samples from this stochastic process as $x_t(x_0, x_1) = \mu_t + \Sigma_t \epsilon$ with $\epsilon \sim \mathcal{N}(0,1)$ and $(x_0, x_1)$, a pair of

reco-level and particle-level events. Moreover, the score $\nabla \log \widehat{\Psi}$ can be learned by minimizing the loss

$$\mathcal{L}_{\text{SB}} = \left\langle \left[ \epsilon_\theta(x_t(x_0, x_1), t) - \frac{x_t(x_0, x_1) - x_0}{\sigma_t} \right]^2 \right\rangle_{t \sim \mathcal{U}([0,1]), (x_0, x_1) \sim p(x_{\text{part}}, x_{\text{reco}})}, \qquad (13)$$

where $x_t$ is sampled according to Eq.(12). After training, the network unfolds by numerically solving the reverse SDE Eq.(8) with the $x_{\text{reco}}$ values as the initial conditions.

We follow a slight variation, where the dynamics are reduced to a deterministic process [52]. This can be achieved by replacing the posterior distribution Eq.(12) by its mean and training the network to encode not the score function, but the velocity field of the reverse process, which then takes the form of an ordinary differential equation:

$$dx_t = v_t(x_t|x_0)dt = \frac{\beta_t}{\sigma_t^2}(x_t - x_0)dt. \qquad (14)$$

For the noise schedule, we follow Ref. [26] and use $g(t) = \sqrt{\beta(t)}$, with $\beta(t)$ the triangular function

$$\beta(t) = \begin{cases} \beta_0 + 2(\beta_1 - \beta_0)t, & 0 \le t < \tfrac{1}{2}, \\ \beta_1 - 2(\beta_1 - \beta_0)\left(t - \tfrac{1}{2}\right), & \tfrac{1}{2} \le t \le 1, \end{cases} \qquad (15)$$

with $\beta_0 = 10^{-5}$ and $\beta_1 = 10^{-4}$.

### 2.2.2 Direct diffusion

Like the Schrödinger Bridge, Direct Diffusion (DiDi) describes a time evolution between particle-level events at $t = 0$ and reco-level events at $t = 1$. Following the Conditional Flow Matching (CFM) [53] framework, DiDi uses an ordinary differential equation (ODE)

$$\frac{dx(t)}{dt} = v_\theta(x(t), t), \qquad (16)$$

with a velocity field $v_\theta(x(t), t)$ encoded in a neural network. This time evolution of the individual events is related to the time evolution of the underlying density via the continuity equation

$$\frac{\partial p(x,t)}{\partial t} + \nabla_x[p(x,t)v_\theta(x,t)] = 0. \qquad (17)$$

The learning task is then to find a velocity field that transforms the density $p(x,t)$ such that

$$p(x,t) \rightarrow \begin{cases} p_{\text{gen}}(x), & t \to 0, \\ p_{\text{sim}}(x), & t \to 1. \end{cases} \qquad (18)$$

Such a velocity field can be constructed by building on event-conditional velocity fields. For a given particle-level event $x_0 \sim p_{\text{gen}}(x_{\text{part}})$, the algorithm samples a corresponding reco-level event $x_1 \sim p_{\text{sim}}(x_{\text{reco}}|x_{\text{part}} = x_0)$, and the two are connected with a linear trajectory

$$x(t|x_0) = (1-t)x_0 + tx_1 \rightarrow \begin{cases} x_0, & t \to 0, \\ x_1 \sim p(x_{\text{reco}}|x_{\text{part}} = x_0), & t \to 1. \end{cases} \qquad (19)$$

Differentiating this trajectory defines the conditional velocity field

$$v(x(t|x_0), t|x_0) = \frac{d}{dt}[(1-t)x_0 + tx_1] = -x_0 + x_1. \qquad (20)$$

This is not yet useful as an unfolding network, as it can only unfold to a pre-specified hard event. The desired unconditional velocity field can be obtained via

$$v(x,t) = \int dx_0 \; \frac{v(x,t|x_0)p(x,t|x_0)p_{\text{gen}}(x_0)}{p(x,t)} \,, \tag{21}$$

where $p(x,t|x_0)$ is the conditional density defined via sampling from equation (19) and $p(x,t)$ is obtained by integrating out the condition $x_0$. In practice, it is sufficient to train on fixed data pairs $(x_{\text{part}}, x_{\text{reco}})$ instead of resampling the posterior $p(x_{\text{reco}}|x_{\text{part}} = x_0)$ in each epoch. The velocity field can be learned from data as a simple regression task with the MSE loss

$$\mathcal{L}_{\text{DiDi}} = \Big\langle [v_\theta((1-t)x_0 + tx_1, t) - (x_1 - x_0)]^2 \Big\rangle_{t \sim \mathcal{U}([0,1]),(x_0,x_1) \sim p(x_{\text{part}}, x_{\text{reco}})}. \tag{22}$$

Once the network is trained, a reco-level event $x_1 \sim p(x_{\text{reco}})$ can be transferred by numerically solving the coresponding ODE in Eq.(16)

$$x_0 = x_1 - \int_0^1 v_\theta(x(t),t)dt. \tag{23}$$

**Unpaired DiDi**    The starting premise of most unfolding methods is that the forward model $p(x_{\text{reco}}|x_{\text{part}})$ is known, within uncertainty. There may be cases where it is not known [25] and instead of pairs $(x_{\text{part}}, x_{\text{reco}})$, we only have access to the marginals $\{x_{\text{part}}\}, \{x_{\text{reco}}\}$. There is no unique solution to this problem even if the detector response is deterministic; however, we can proceed by assuming that the function corresponds to the optimal transport map. We consider a variation of DiDi for this configuration by dropping the pairing information between training events [27]. This can be achieved by modifying the conditional trajectory so that $x_1$ is sampled independently of $x_0$, so Eq.(19) becomes

$$x(t|x_0) = (1-t)x_0 + tx_1 \rightarrow \begin{cases} x_0, & t \rightarrow 0, \\ x_1 \sim p(x_{\text{reco}}), & t \rightarrow 1. \end{cases} \tag{24}$$

The loss function is

$$\mathcal{L}_{\text{DiDi-U}} = \Big\langle [v_\theta((1-t)x_0 + tx_1, t) - (x_1 - x_0)]^2 \Big\rangle_{t \sim \mathcal{U}([0,1]), x_0 \sim p(x_{\text{part}}), x_1 \sim p(x_{\text{reco}})}. \tag{25}$$

During training we now sample events independently of each other, and the learned map will be purely determined by the network and its training.

**Bayesian network**    Because the distribution mapping loss function does not have a straightforward interpretation as a likelihood, it cannot be simply transformed into a Bayesian network from first principles. However, we can add the relevant features of a Bayesian network, as for the CFM [27,47]. This includes Bayesian layers, Gaussian distributions of all or some network parameters, and a KL-term regularizing the network parameters towards a Gaussian prior,

$$\mathcal{L}_{\text{B-CFM}} = \Big\langle \mathcal{L}_{\text{CFM}} \Big\rangle_{\theta \sim q(\theta)} + c\text{KL}[q(\theta), p(\theta)]. \tag{26}$$

The factor $c$ balances the deterministic loss with the Bayesian-inspired regularization. If the network loss follows from a likelihood, this factor is fixed by Bayes' theorem. In all other cases it is a hyperparameter. We have checked that the network performance as well as the extracted posteriors are stable when varying $c$ over several orders of magnitudes, suggesting that the learned weight distribution corresponds to an inherent property of the setup.

## 2.3 Generative unfolding: cINN, Transfermer, CFM, TraCFM, Latent Diffusion

Generative unfolding uses conditional generative networks to learn the conditional probability describing the inverse simulation $p_{\text{model}}(x_{\text{part}}|x_{\text{reco}})$,

$$
\begin{array}{ccc}
p_{\text{gen}} & & p_{\text{unfold}}(x_{\text{part}}) \\
\text{paired data} \updownarrow & & \uparrow p_{\text{model}}(x_{\text{part}}|x_{\text{reco}}) \\
p_{\text{sim}} & \xleftrightarrow{\text{correspondence}} & p_{\text{data}}(x_{\text{reco}})
\end{array}
\tag{27}
$$

Building a forward surrogate network for $p(x_{\text{reco}}|x_{\text{part}})$ uses the same data and has nearly the same setup as learning the inverse probability $p(x_{\text{part}}|x_{\text{reco}})$. The usual assumption of unfolding is that the detector response is universal, which breaks the symmetry of the forward and backwards networks via Bayes' theorem,

$$
p(x_{\text{part}}|x_{\text{reco}}) = p(x_{\text{reco}}|x_{\text{part}}) \frac{p(x_{\text{part}})}{p(x_{\text{reco}})}.
\tag{28}
$$

For the forward simulation, we assume that the condition on $x_{\text{part}}$ does not induce a significant prior for the generated $p_{\text{sim}}$. For the inverse simulation, this prior dependence is relevant and it formally implies that there is no notion of unfolding single events, even though the generative unfolding tools provide the corresponding conditional probabilities.

Technically, we start from a simple latent distribution, where the generative network transforms the required phase space distribution,

$$
z \sim p_{\text{latent}}(z) \quad \xrightarrow{G_\theta(z;x_{\text{reco}})} \quad x_{\text{part}} \sim p_{\text{model}}(x_{\text{part}}|x_{\text{reco}}).
\tag{29}
$$

The phase space distribution of an unfolded dataset is then given as

$$
p_{\text{unfold}}(x_{\text{part}}) = \int dx_{\text{reco}} \, p_{\text{model}}(x_{\text{part}}|x_{\text{reco}}) \, p_{\text{data}}(x_{\text{reco}}).
\tag{30}
$$

This approach is based on posterior distributions for individual events, which means that we can also take single measured events and run them through the model any number of times. In practice, Eq. 30 is achieved by sampling from $p_{\text{model}}(x_{\text{part}}|x_{\text{reco}})$ for data events that follow $p_{\text{data}}(x_{\text{reco}})$ and then ignoring the $x_{\text{reco}}$ argument from the resulting dataset of pairs $(x_{\text{part}}, x_{\text{reco}})$. If we wanted to sample further from the result and/or iterate the procedure, we would need to do something like the second step of OmniFold, which is a local averaging as done for generative models in Ref. [31]. A key ingredient to unfolding with generative networks [28] is to either train this network with a likelihood loss [29], like for the cINN, or to guarantee the probabilistic interpretation through the mathematical setup, like in the CFM.

### 2.3.1 Conditional INN

The original generative network used for unfolding is a normalizing flow [54] in its conditional invertible neural network (cINN) variant [55, 56]. It defines the mapping between the latent and phase space as an invertible function, conditioned on the reco-level event,

$$
z \sim p_{\text{latent}}(z) \quad \underset{\leftarrow G_\theta^{-1}(x_{\text{part}};x_{\text{reco}})}{\overset{G_\theta(z;x_{\text{reco}}) \rightarrow}{\longleftrightarrow}} \quad x_{\text{part}} \sim p_{\text{model}}(x_{\text{part}}|x_{\text{reco}}).
\tag{31}
$$

The bijection form allows us to write down the learned density as

$$p_{\text{model}}(x_{\text{part}}|x_{\text{reco}}) = p_{\text{latent}}(G_\theta^{-1}(x_{\text{part}};x_{\text{reco}})) \left| \det \frac{\partial G_\theta^{-1}(x_{\text{part}};x_{\text{reco}})}{\partial x_{\text{part}}} \right|. \tag{32}$$

Having access to the network likelihood enables us to use it directly as loss function and train via maximum likelihood estimation

$$\mathcal{L}_{\text{cINN}} = -\left\langle \log p_{\text{model}}(x_{\text{part}}|x_{\text{reco}}) \right\rangle_{(x_0,x_1)\sim p(x_{\text{part}},x_{\text{reco}})}. \tag{33}$$

This approach requires a bijective map that is flexible enough to model complex transformations, while still allowing for efficient computation of the Jacobian determinant. We employ coupling blocks [56], but replace the affine coupling blocks with the more flexible rational quadratic spline blocks [57].

**Transformer-cINN**    We also consider a transformer extension to the standard cINN [58]. The architecture translates a sequence of reco-level momenta into a sequence of particle-level momenta. A transformer network encodes the correlations between all event dimensions at particle level as well as their correlation with the reco-level event. A small 1D-cINN then generates the hard-level momenta conditioned on the transformer output. To guarantee invertibility and a tractable Jacobian, the likelihood and the generation process are factorized autoregressively

$$p_{\text{model}}(x_{\text{part}}|x_{\text{reco}}) = \prod_{i=1}^{n} p_{\text{model}}(x_{\text{part}}^{(i)}|c(x_{\text{part}}^{(0)},\dots,x_{\text{part}}^{(i-1)},x_{\text{reco}})). \tag{34}$$

The product in Eq.(34) covers all dimensions at particle level. The function $c$ is learned by the transformer to encode the information about the reco-level momenta as well as the already generated hard-level momenta. The one-dimensional conditional densities are encoded in the cINN. Note that in contrast to Ref. [58], this so-called transfermer is autoregressive in individual one-dimensional components, instead of in the four momenta grouped by particles.

### 2.3.2   Conditional flow matching

As an alternative generative network, we employ a diffusion approach called Conditional Flow Matching (CFM) [47, 53]. The mathematical structure is the same as for the DiDi network introduced in Sec. 2.2.2. The key difference here is that the CFM now samples from a Gaussian latent distribution, conditional on a reco-level event, Eq.(29). This means the time-evolving density is conditional and interpolates between the boundary conditions

$$p(x,t|x_{\text{reco}}) \rightarrow \begin{cases} p(x_{\text{part}}|x_{\text{reco}}), & t \rightarrow 0, \\ \mathcal{N}(x;0,1), & t \rightarrow 1, \end{cases} \tag{35}$$

while the ODE now reads

$$\frac{dx(t)}{dt} \equiv v_\theta(x(t),t|x_{\text{reco}}). \tag{36}$$

The information about the reco-level event to unfold is no longer encoded in the initial condition of the ODE, but in an additional input to the network that predicts the velocity field. Again we start with paired training data, $x_0 \sim p(x_{\text{part}})$ and $x_1 \sim p(x_{\text{reco}}|x_{\text{part}} = x_0)$, and define a simple conditional trajectory towards Gaussian noise,

$$x(t|x_0,x_{\text{reco}}) = (1-t)x_0 + t\epsilon \rightarrow \begin{cases} x_0, & t \rightarrow 0, \\ \epsilon \sim \mathcal{N}(0,1), & t \rightarrow 1. \end{cases} \tag{37}$$

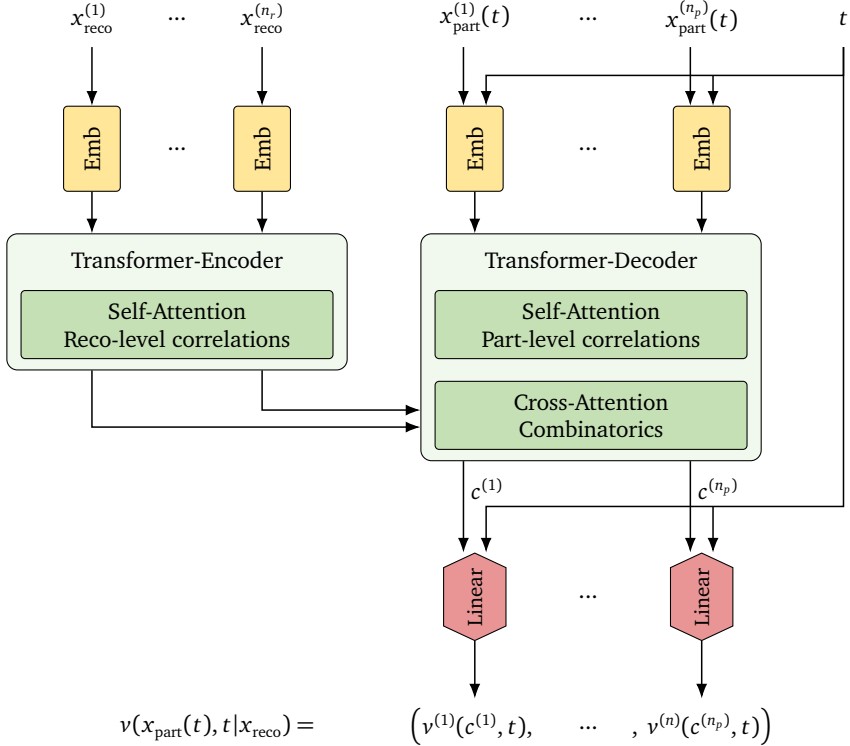

Figure 1: TraCFM architecture, combining the CFM generator with a Transformer encoder-decoder combination to improve combinatorics.

The conditional velocity field is defined via the derivative of the trajectory

$$v(x(t|x_0, x_{\text{reco}}), t|x_0, x_{\text{reco}}) = \frac{d}{dt}\left[(1-t)x_0 + t\epsilon\right] = -x_0 + \epsilon. \tag{38}$$

The rest of the derivation follows analogously to the DiDi derivation. The loss function is given by the MSE

$$\mathcal{L}_{\text{CFM}} = \left\langle \left[v_\theta((1-t)x_0 + t\epsilon, t, x_1) - (\epsilon - x_0)\right]^2 \right\rangle_{t \sim \mathcal{U}([0,1]), (x_0, x_1) \sim p(x_{\text{part}}, x_{\text{reco}}), \epsilon \sim \mathcal{N}}. \tag{39}$$

After training, the CFM can unfold by sampling from the latent noise distribution and solving the ODE in Eq.(36) conditioned on the reco-level event we want to unfold. The crucial difference to DiDi is that this procedure allows us to unfold the same reco-level event repeatedly, each time from different noise as starting point, to sample the posterior distribution $p_{\text{model}}(x_{\text{part}}|x_{\text{reco}})$.

**Transformer-CFM** The velocity field can be encoded in any type of neural network, in that sense CFMs do not impose any architectural constraints. While linear layers already achieve high precision [27, 47], we find that when dealing with complex correlations employing a Transformer network further improves results [58], similar to the INN vs Transfermer case.

Our TraCFM architecture encoding $v(x_{\text{part}}, t|x_{\text{reco}})$ is shown in Fig. 1. Its inputs are the reco-level event, the intermediate noisy diffusion state $x_{\text{part}}(t)$ and the time $t$. First, each of the reco-level and particle-level dimensions is individually mapped into a higher-dimensional embedding space. This is done by concatenating the kinematic variable with its one-hot-encoded position and filling with zeros up to the specified embedding dimension [58]. For the particle-level dimensions we also concatenate the time $t$ to the vector before filling with zeros.

We experimented with more sophisticated embedding strategies, but found no performance improvements. The reco-level embeddings are then fed to the transformer encoder, which encodes the correlations among them using a self-attention mechanism. The transformer decoder does the same for the particle-level dimensions. Finally, the updated embeddings are fed to a cross-attention block that learns to resolve the combinatorics between reco-level and particle-level objects and outputs a final condition $c^{(i)}$ for each particle-level dimension. A single linear layer, shared between all dimensions, maps this condition together with the time $t$ to the individual velocity field components.

To unfold, we start with a sample from the latent distribution as $x_{\text{part}}(t=1) = \epsilon \sim \mathcal{N}(0,1)$ and solve the ODE Eq. (36) numerically. Notice that the transformer encoder has no time dependence, so we do not need to recalculate it at every function call.

**Bayesian generative network** The concept of Bayesian networks can be applied to generative networks by assigning an uncertainty to the learned underlying phase space density. This way, the network learns an underlying density to sample from, and an uncertainty on this density which it can report as an error of the unit-weight of each generated event [45, 46]. Because the loss of the normalizing flow is a maximized likelihood, the relation between the likelihood loss and the regularizing KL-divergence can be derived from Bayes' theorem. As an approximation to the full posterior, the error bars reported by Bayesian networks are a learned approximation to the true uncertainty on the phase space density.

### 2.3.3 Latent variational diffusion

To reduce the disparity between different parameterizations of the set of observables and to enable a more robust network, Latent Variational Diffusion [34] introduces a Variational Autoencoder to initially map observables from particle/parton phase space to a latent space. This particle encoder learns the mapping

$$x_{\text{part}} \rightarrow z = \text{ENCODER}_{\text{part}}(x_{\text{part}}) \in \mathbb{R}^{D_{\text{Latent}}}. \tag{40}$$

It is implemented as a deep feed-forward network. This latent space can be fine-tuned for the diffusion step, allowing enhanced control over the generation process before mapping the result back to the observables.

To accommodate variable-length reco-level objects, an additional detector encoder maps them to a fixed-length latent vector

$$x_{\text{reco}} \rightarrow w = \text{ENCODER}_{\text{reco}}(x_{\text{reco}}) \in \mathbb{R}^{D_{\text{Latent}}}. \tag{41}$$

It utilizes a deep feed-forward network for fixed-length inputs and a transformer encoder for variable-length inputs.

These latent observables provide the inputs for a conditional variational diffusion network [59]. VLD employs a continuous-time, variance-preserving, stochastic diffusion process with a noise-prediction parameterization for the score function. The governing stochastic differential equations are

$$dz = f(z_t, t)\,dt + g(t)\,dw \qquad \text{(VLD Forward SDE)},$$
$$dz = \left[f(z_t, t) - g^2(t)s_\theta(z, w, t)\right]dt + d\bar{w} \qquad \text{(VLD Reverse SDE)}, \tag{42}$$

where $f$ is the drift term of the SDE, $g$ is the diffusion coefficient, and $s$ is the score function, as in Eqs. (6) and (7). The drift and diffusion terms are parameterized through a learnable noise

schedule $\gamma_\phi(t)$, which controls the diffusion rate. It is encoded in a monotonically increasing deep network as a function of $t$,

$$
f(z, t) = -\frac{1}{2}\left[\frac{d}{dt}\log\left(1 + e^{\gamma_\phi(t)}\right)\right]z,
$$
$$
g(t) = \sqrt{\frac{d}{dt}\log\left(1 + e^{\gamma_\phi(t)}\right)}. \tag{43}
$$

This simplifies the forward process to a time-dependent normal distribution, controlled by $\gamma_\phi(t)$, now interpreted as the logarithmic signal-to-noise ratio.

$$
z_t \sim \mathcal{N}\left(\sigma(-\gamma_\phi(t))z, \sigma(\gamma_\phi(t))\mathbb{I}\right), \qquad \text{where} \quad \sigma(x) = \sqrt{\frac{1}{1 + e^{-x}}}. \tag{44}
$$

The diffusion score is parameterized via a noise-prediction network,

$$
s_\theta(z, w, t) = \frac{\hat{\epsilon}_\theta(z_t, w, t)}{\sigma(-\gamma_\phi(t))}, \tag{45}
$$

trained to predict the sampled noise used to generate the forward sample from the diffusion process [34, 59]. It is implemented as a deep feed-forward network, concatenating the three inputs before processing.

Finally, a decoder transforms the initial noisy latent particle representation back into phase space observables,

$$
z_0 \; \rightarrow \; \hat{x}_{\text{part}} = \text{Decoder}(z_0). \tag{46}
$$

It is again implemented as a deep feed-forward network and outputs real-valued estimates of the observables.

All networks are trained in a end-to-end fashion using a unified loss which allows the encoders and decoders to fine-tune the latent space to the diffusion process, while accurately reconstructing the observables. We use a standard normal distribution as the prior over the final noisy latent vector, $p(z_1) \sim \mathcal{N}(\mathbf{0}, \mathbb{I})$. The denoising network, $\gamma_\phi(t)$, is trained to minimize the variance of the following loss term, while all other networks are trained to minimize its expectation value:

$$
\begin{aligned}
\mathcal{L}_{\text{VLD}} = {} & \text{KL}[q(z_1|x), p(z_1)] & \text{(Prior Loss)} \\
& + \left\langle \|\text{Decoder}(z_0) - x\|_2^2 \right\rangle_{q(z_0|x)} & \text{(Reconstruction Loss)} \\
& + \left\langle \gamma_\phi'(t)\|\epsilon - \hat{\epsilon}_\theta(z, w, t)\|_2^2 \right\rangle_{\epsilon \sim \mathcal{N}(\mathbf{0}, \mathbb{I}), t \sim \mathcal{U}(0,1)} & \text{(Denoising Loss).} \tag{47}
\end{aligned}
$$

## 3  Detector unfolding: $Z$+jets

### 3.1  Data and preprocessing

As a first test case for the various ML-Unfolding methods we use a new, bigger version of the public dataset from Ref. [15], now available at Ref. [60].[1] The events describe

$$
pp \rightarrow Z + \text{jets}, \tag{48}
$$

production at $\sqrt{s} = 14$ TeV, simulated with Pythia 8.244 [62] with Tune 26. In contrast to the original dataset, detector effects are now simulated with the updated Delphes 3.5.0 [63],

---

[1]For a comparison with classical unfolding methods, we refer to Refs. [15] and [61].



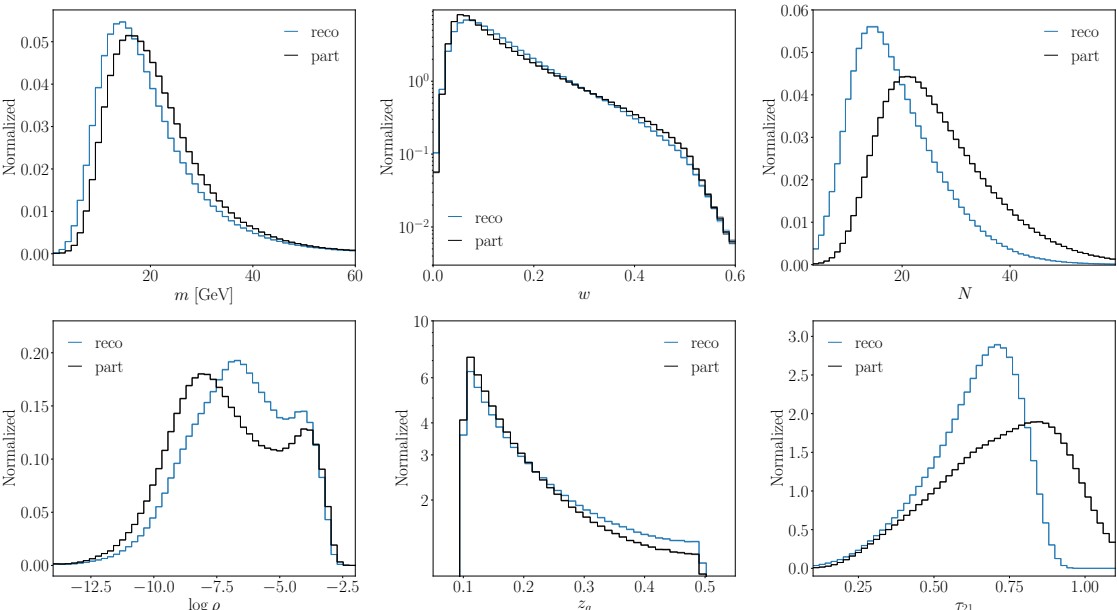

Figure 2: Subjet distributions for the $Z$+jets dataset, at the particle level, and at the reco level.

and the CMS tune, that uses particle flow reconstruction. The jets are clustered using all particle flow objects available at detector level and all stable non-neutrino truth particles at particle level. Jets are defined by the anti-$k_T$ algorithm [64] with $R = 0.4$, as implemented in FastJet 3.3.2 [65]. The dataset contains around 24M simulated events, 20M for training and 4M for testing.

We focus on six observables describing the leading jet: mass $m$, width $\tau_1^{(\beta=1)}$, multiplicity $N$, soft-drop [66, 67] mass $\rho = m_{\text{SD}}^2/p_T^2$ and momentum fraction $z_g$ using $z_{\text{cut}} = 0.1$ and $\beta = 0$, and the $N$-subjettiness ratio $\tau_{21} = \tau_2^{(\beta=1)}/\tau_1^{(\beta=1)}$ [68]. For 0.8% of the events we map an undefined jet groomed mass $\log \rho$ or N-subjettiness ratio $\tau_{21}$ to zero.

The distributions are shown in Fig. 2. We apply a dedicated preprocessing to the jet multiplicity and the groomed momentum fraction. The jet multiplicity is an integer feature, which forces the network to interpolate, so we smooth the distribution by adding uniform noise $u \sim \mathcal{U}[-0.5, 0.5]$. This preprocessing can be inverted. The groomed momentum fraction features a discrete peak at $z_g = 0$ and sharp cuts at $z_g = 0.1$ and 0.5. We move the peak to $z_g = 0.097$ and add uniform noise $u \sim \mathcal{U}[0, 0.003]$. Next, we take the logarithm to make the distribution more uniform. We then shift and scale the distribution to stretch from $-1$ to $+1$ and take the inverse error function to transform its shape to an approximate normal distribution. Finally, all six observables are standardized by subtracting the means and dividing by the standard deviations.

Also in Fig. 2, we show the effect of the detector simulation. They are most significant for the jet multiplicity, the groomed jet mass, and the $N$-subjettiness ratio. All these shifts are driven by the finite energy threshold of the detector.

## 3.2 Reweighting

As in Sec. 2, we start with the OmniFold reweighting on the $Z$+jets dataset. We then introduce the Bayesian version (bOmniFold) and compare their performance. We train both networks for two different unfolding tasks. First, we evaluate their performance on the same dataset as the other generative networks, splitting it in two halves, but adding noise to one of them as

described below. Then, we go back to the previous Pythia dataset and task the classifiers with learning the likelihood ratio between Pythia and Herwig. We use this ratio to reweight Herwig onto Pythia.

### 3.2.1 Training on Pythia with added noise

For this section, we employ the combination of Pythia with the updated Delphes version. We merge the training and test sets with 24.3M events, of which we use 10.9M for training, 1.2M for validation, and 12.2M for testing. In each of these splits, we label half of the events as Pythia 1 and the other half as Pythia 2. The classifier has to learn to reweight Pythia 1 onto Pythia 2.

If we train (b)Omnifold on this task, it will just learn a constant classifier value of 0.5, so we add Gaussian noise $\varepsilon \sim \mathcal{N}(0,1)$ to each of the raw features before preprocessing, scaled by the standard deviation of the respective feature $\sigma_x$ and an additional custom factor $f$ to modify the relative importance of the noise,

$$x \to \tilde{x} = x + f \cdot \sigma_x \varepsilon, \qquad \text{with} \qquad \varepsilon \sim \mathcal{N}(0,1), \tag{49}$$

where we use $f = 0.1$.

We train OmniFold (13k parameters) and its Bayesian-network counter part bOmniFold (2×13k parameters) with identical settings for 30 epochs. The unfolded distributions are shown in Fig. 3. While this reweighting task might not be realistic, it defines an illustrative benchmark for the performance of an unfolding network. For each of the one-dimensional kinematic distributions, the agreement between the unfolded and true particle-level events is at the percent level over most of the phase space. The only exceptions are sparsely populated tails with too little training data, or sharp features with limited resolution. The differences between the OmniFold and bOmniFold results are even smaller. A selection of summary statistics are presented in Tab. 1, where we show the Wasserstein 1-distance, the triangular distance,

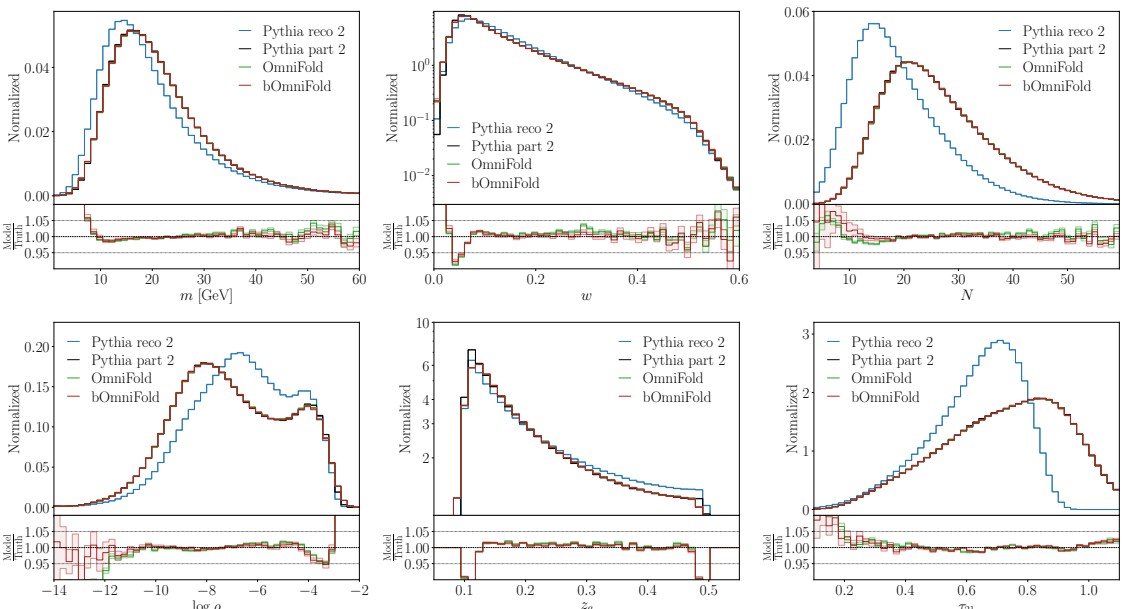

Figure 3: Unfolded distributions from event reweighting using OmniFold and bOmniFold. The bOmniFold error bar is based on drawing 20 Bayesian samples. For OmniFold the error bar represents the bin-wise statistical uncertainty.

Table 1: Metrics evaluating the performance of the different unfolding networks, for each of the one-dimensional kinematic distributions. We show the Wasserstein 1-distance ($\times 10$), the triangular distance ($\times 1000$), and the energy distance ($\times 1000$).

|  | $m$ [GeV] | $w$ | $N$ |
|---|---|---|---|
| OmniFold | 0.59098 / **0.12493** / 13.72203 | 0.01001 / **1.62601** / 2.99618 | 0.67919 / 0.03034 / 18.47942 |
| bOmniFold | **0.37180** / 0.14208 / **9.89718** | **0.00542** / 1.64286 / **2.24587** | **0.22693** / **0.02176** / **4.97982** |

|  | $\log \rho$ | $z_g$ | $\tau_{21}$ |
|---|---|---|---|
| OmniFold | 0.40320 / 0.72494 / 15.60005 | 0.01550 / 15.30356 / 4.81947 | **0.00931** / 0.02746 / **1.40143** |
| bOmniFold | **0.12501** / **0.67605** / **5.59003** | **0.01109** / **15.27470** / **4.51572** | 0.00956 / **0.02183** / 1.54405 |

and the energy distance for the six kinematic observables. The two methods were not separately optimized, we just started with a generic OmniFold setup and supplemented it with the Bayesian network features. Uncertainties on the statistics are not included in these illustrative metrics.

For the uncertainties, we see that it tends to cover the deviation of the unfolded distributions from the truth target towards increasingly sparse tails. Far in the tails, where there is too little training data altogether, the networks learn neither the density nor an error bar on it.

### 3.2.2 Reweighting Herwig onto Pythia

For a more realistic (b)OmniFold task, we go back to the original Pythia dataset, for which we also have a Herwig [69] version with the same version of Delphes, as introduced in Ref. [15]. We train (b)OmniFold for 500 epochs on 2M events, and test on 664k events [26].

First, we show the losses as a function of the training in Fig 4. This comparison shows the challenge of the classifier training, which rapidly overtrains after about 20 epochs. This behavior does not appear in the previous study with noisy Pythia events and is due to the smaller training data for the Herwig reweighting. For large numbers of training epochs, the loss on the validation dataset indicates a decreasing performance due to overtraining. For applications which require an LHC-level of precision, such overtraining may become a problem. It can be avoided, for instance, using regularization techniques, such as dropout. Both of these mechanisms are part of the Bayesian network architecture, in case of the regularization with

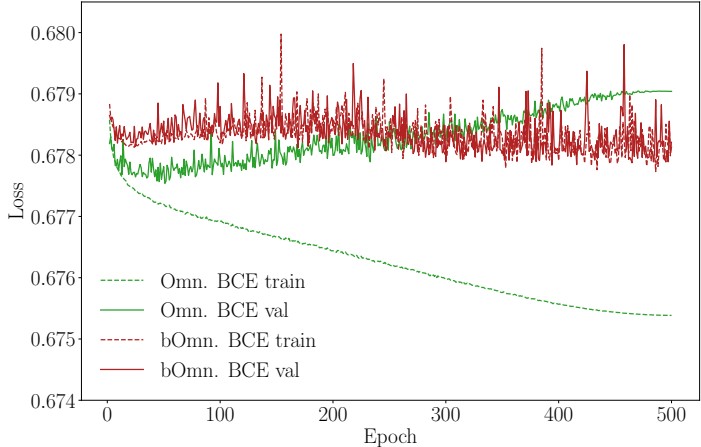

Figure 4: BCE losses during training for 500 epochs for Omnifold (green) and bOmnifold (red), for Herwig-to-Pythia reweighting.

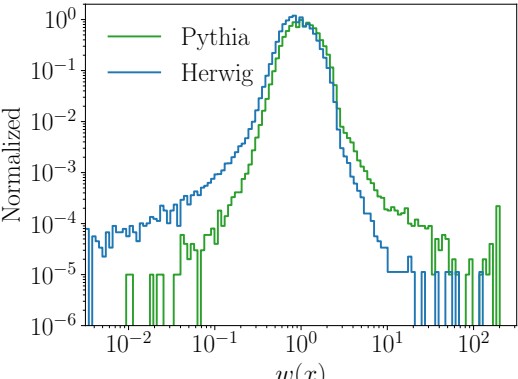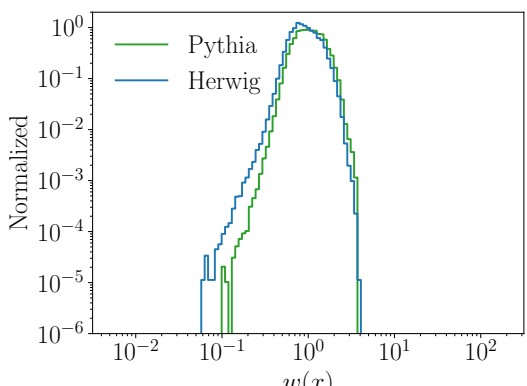

Figure 5: Weight distribution (clipped at 200) in the training set for Herwig-to-Pythia reweighting: OmniFold (left) vs bOmniFold (right). For each network we histogram the weights for the Herwig and Pythia data points.

a strength given by Bayes' theorem. In Fig. 4 we see that the bOmniFold training continues to improve even after a large number of epochs, with no overtraining. Interestingly, bOmniFold has larger epoch-to-epoch fluctuations and has a worse minimum validation loss than Omni-Fold, but does not show signs of overtraining. This illustrates a potential tradeoff between accuracy and stability.

We verified that the unfolded observables between OmniFold and bOmbiFold are in agreement. Because of the difference between the training data, or prior, and the data we then unfold, the true particle-level distributions are not exactly reproduced. The interesting feature of the bOmniFold training is that it has suppressed tails in the weight distribution with respect to OmniFold, as shown in Fig. 5, even though both networks learn the same reweighting map. Large and small weights lead to undesired statistical dilution of the dataset, and it will be interesting to explore in the future the interplay between statistical dilution and accuracy.

## 3.3   Mapping distributions

The same subjet unfolding can be tackled with distribution mapping, using the Schrödinger Bridge and Direct Diffusion, both introduced in Sec. 2.2.1. The implementation of the Schrödinger Bridge follows the original Pytorch [70] implementation [26]. The noise prediction network is implemented using a fully connected architecture with additional skip connections, specifically using six RESNET [71] blocks, with each residual layer connected to the output of a single MLP through a skip connection. The Bayesian version replaces the original MLPs. The training uses the Adam [72] optimizer. The total number of trainable parameters is around 2M split equally between the mean and standard deviation of the trainable weights.

During data generation, we sample using the MAP prediction, i.e. fix every network weight at the learned mean. Uncertainties are derived by sampling 50 times from the learned weight distributions. In Fig. 6, we quantify the agreement between the unfolded and truth one-dimensional kinematic distributions. The unfolding performance can be compared to the noisy reweighting benchmark in Fig. 3. The agreement between unfolded and truth-level observables is still precise to the percent level. Notably, the largest deviations from the truth distribution occur in the low-statistics edges, while the bulks of the distributions are well described by the generative mapping, and the deviations from the truth are well covered by the Bayesian uncertainty.

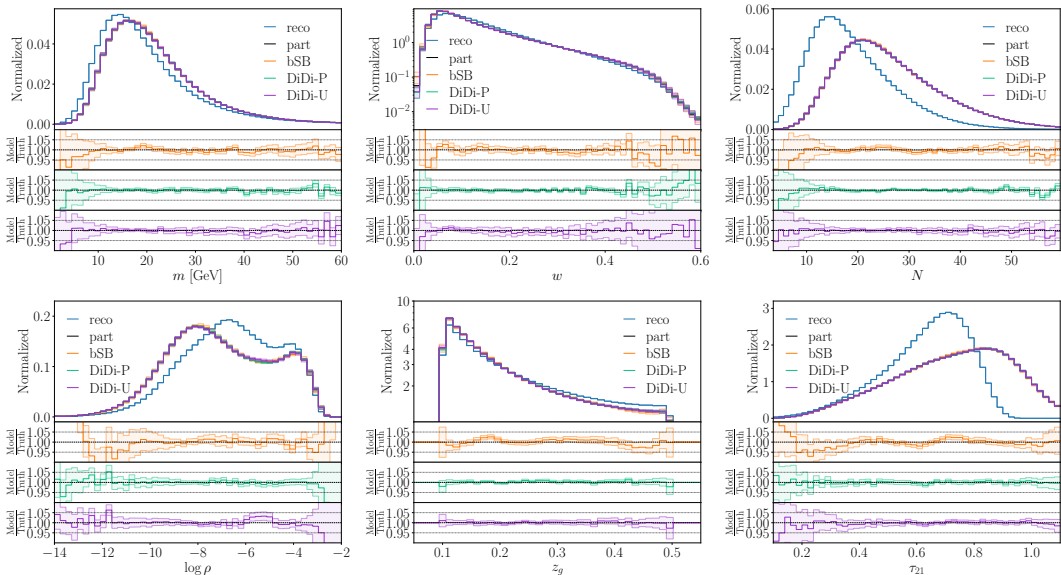

Figure 6: Unfolded distributions from distribution mapping, using the Schrödinger Bridge and DiDi. The Bayesian error bars are based on drawing 50 samples.

An alternative method for the same tasks is Direct Diffusion. We encode the velocity field in a standard Bayesian MLP, after not seeing better results with more advanced networks. Again, we implement the network in Pytorch, train it using the Adam optimizer, use the MAP prediction, and draw 50 Bayesian weight samples to estimate the uncertainties. We use the same setup for paired and unpaired DiDi. The only difference is the reshuffling of the reco-particle pairings at the beginning of each epoch in the unpaired setting. The network size comes to about 3M parameters, 1.5M weights each associated with the mean and the standard deviation.

The results are compared to the Schrödinger Bridge in Fig. 6. Both variants of Direct Diffusion learn the observables with percent precision over the entire phase space, and better than that in the well-sampled bulk. For the central prediction, the paired training data makes the unfolding slightly more precise and more stable.

A difference between paired and unpaired DiDi is that the latter might be slightly less stable and learns significantly larger Bayesian uncertainties. This is consistent over several trainings. At the level of kinematic distribution we do not observe any shortcoming for unfolding through distribution matching, and the difference between paired and unpaired training data is minor. We will come back to the conceptual difference in Sec. 3.5.

## 3.4 Generative unfolding

The third unfolding method we study is based on learned conditional probabilities, as defined in the statistical description of unfolding. It relies on paired training data. Differences appear when we vary the generative network architecture used. We skip the original GAN implementation [28], because more modern generative networks have been shown to learn phase space distributions more precisely [46, 47]. The cINN [29] is implemented in PyTorch [70] and uses the FrEIA library [73] with RQS coupling blocks. By default, we use its Bayesian version [45], which tracks the uncertainty in the learned phase space density as variations of unit event weights. Our CFM [47] encodes the velocity field in the same linear layer architecture as DiDi, including the Bayesian version. Predictions are obtained by unfolding each event 30 times with the MAP weights, and the uncertainties are obtained from drawing 50 sets of weights from the Bayesian network.

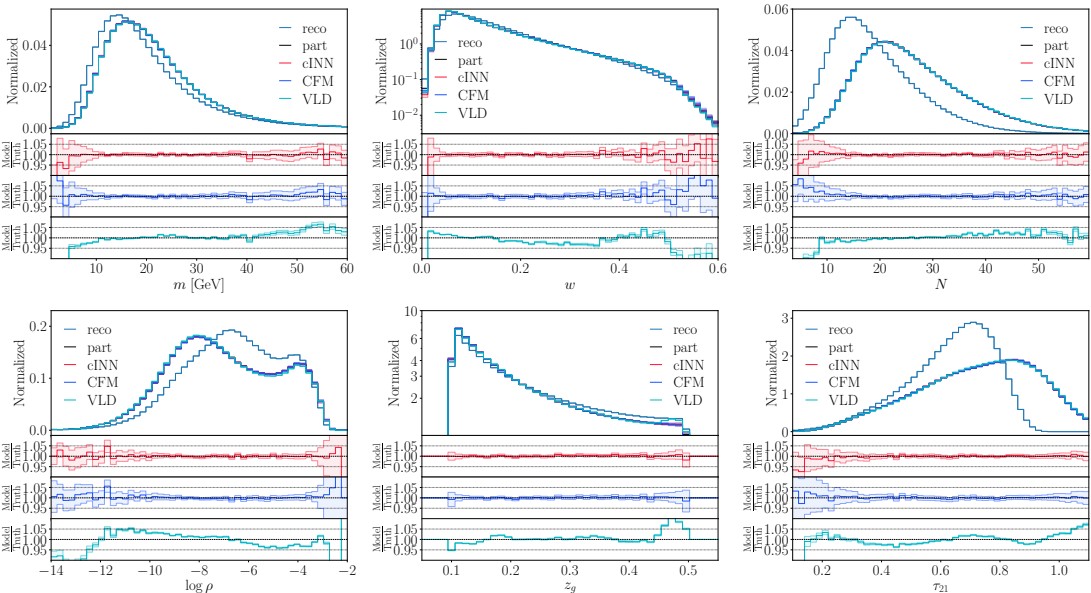

Figure 7: Unfolded distributions from conditional generation, using cINN, CFM and VLD. For cINN and CFM, the Bayesian errors are based on drawing 50 samples, and the MAP estimate is obtained by unfolding each event 30 times. For VLD, we show the bin-wise mean and standard deviation of 33 unfoldings.

VLD is implemented and trained using the same JAX codebase released alongside Ref. [34]. Observables are first pre-processed so that each marginal distribution follows a standard normal via a quantile transform. Predictions are generated using the DPM++ multi-step solver [74] with 1000 inference steps and the learned schedule. Unlike the other models, VLD is not implemented as a Bayesian network. Uncertainties are estimated by sampling each unfolding 33 times using a different seed for generating the prior noise.

In Fig. 7 we show the results from the cINN, CFM, and VLD. As for the (b)OmniFold reweighting and the distribution matching, all kinematic distributions are reproduced at the percent level or better. While the performance of the cINN and the CFM are very similar, the VLD approach shows slightly larger deviations from the target distributions.

## 3.5 Learned event migration

For the generative network methods, it can be instructive to examine the learned map between reco events and truth events. In the top panels of Fig. 8 we start with the migration described by the paired events from the forward detector simulation. We show three of the kinematic distributions defined in Fig. 2. The results for the other distributions lead to the same conclusions. For the jet mass, the multiplicity, and $\tau_{21}$ we see that the optimal transport defined by the detector is quite noisy. While for the jet mass most events form a diagonal with little bias, the jet multiplicity shows a linear correlation with a non-trivial slope, and $\tau_{21}$ indicates a saturation effect in the forward direction $x_{\text{part}} \rightarrow x_{\text{reco}}$, such that $\tau_{21}(x_{\text{reco}})$ does not reach one.

In the next row we show the results from the Schrödinger Bridge, which is similar, but slightly noisier than DiDi trained on paired events, shown in the third row. Using paired events, these generative networks learn a very efficient diagonal transport map, with a spread that is more narrow than the actual detector. The main features of the detector truth are reproduced well.

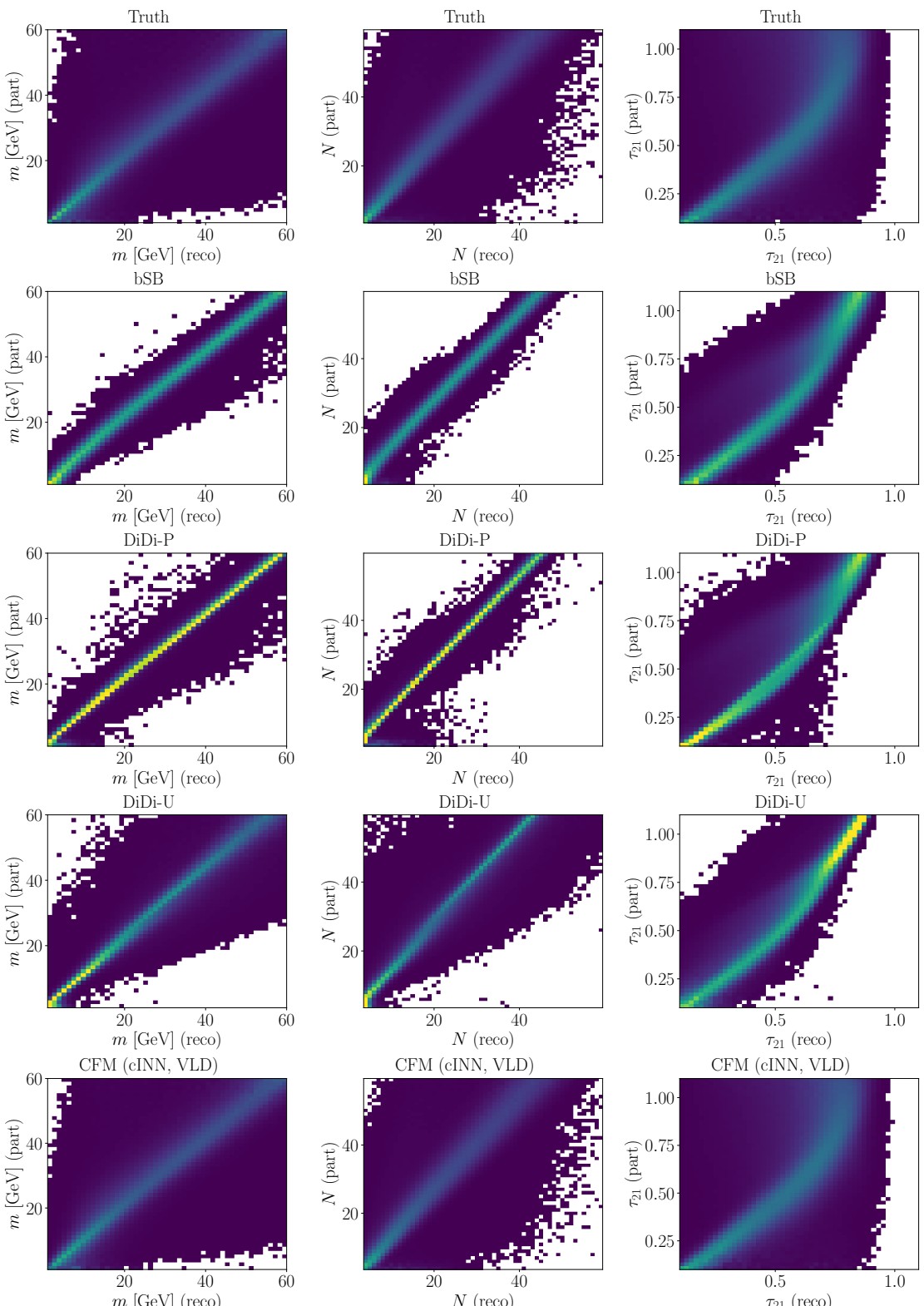

Figure 8: Migration maps for three representative distributions. From the top: forward detector simulation, Schrödinger Bridge, paired DiDi, unpaired DiDi, and CFM/cINN/VLD, which are all looking identical. The bin contents are normalized such that each row sums to one.

Next, we see that the unpaired DiDi network again learns an efficient transport map, but with a significantly broader spread than the same network trained on paired events. The reason is that ignoring the event pairing leads to a noisier training, but again reproducing the main features of the detector. We emphasize that unpaired training seems to bring DiDi-like implementations closer to describing the actual detector, but this is an artifact in that the detector mapping is noisier than the optimal transports from distribution mapping, and training on unpaired samples is also noisier, but the two are not positively related. Finally, we show the transport learned by the conditional CFM networks. Not shown are the corresponding cINN and VLD results, which are visually identical to the CFM results. The conditional generative models indeed learn the correct detector transport from the paired events, indicating that conditional generative networks indeed encode the conditional probabilities from Eq.(27).

## 3.6   Classifier check

Finally, there is the question if the learned distributions have failure modes that cannot be seen from the marginal distributions. Following [75–77], we systematically search for mismatched correlations using a trained binary classifier between the true training events and the same number of unfolded events. Using Eq.(2) this classifier can be turned into a re-weighting function $w(x)$, evaluated for each data point in phase space.

We train individual classifiers for each of the discussed methods, using the hyperparameters listed in Tab. 7. The results are presented in Fig. 9. The top left plot shows the phase space weights $w$ for the three distribution mapping methods bSB, DiDi-P and DiDi-U. For all three networks, the dominant feature is a sharp peak in $w \approx 1$, showing overall excellent agreement between the learned unfolding and the truth distribution. A tail towards higher weights indicates the existence of phase space regions that are underpopulated by the network, while lower weights indicate an overpopulated region. While for all three networks weak tails in both directions exist, they are suppressed by several orders of magnitude compared to the peak around unit weights. The right plot shows the same weight distribution zoomed into the area around $w(x) = 1$ to better visualize the peak. Comparing the three methods, we find a slightly sharper peak for the DiDi networks compared to the bSB implementation.

The bottom row of Fig. 9 shows the same two plots for the three generative networks, cINN, CFM, and VLD. Again, for all three methods almost all events fall into a sharp peak around weight one. Tails towards high and low weights exist, but are again strongly suppressed compared to the peak and only visible due to the logarithmic scale of the $y$-axis. Comparing the three networks, we find a slightly larger spread for the VLD. The right plot shows the zoomed-in version of the same distribution. Here we see that the density in the peak is about one order of magnitude larger for the CFM and cINN, compared to the VLD.

Finally, we can compare the distribution mapping methods in the top row to the generative networks in the bottom row. They show overall very similar performance, indicating that despite their different migration patterns shown in Fig. 8, both model classes can recover the unfolded distribution with high precision. In the high resolution plots on the right, we find a slightly sharper peak around unit weights for the CFM and the cINN, indicating the overall best agreement with the truth distribution, as measured by the trained classifier.

# 4   Unfolding to parton level: Top pairs

## 4.1   Data

As a second benchmark we apply ML-methods to top quark pair production, unfolding from reco-level to parton level, i.e. the level of the top quarks and their decay products

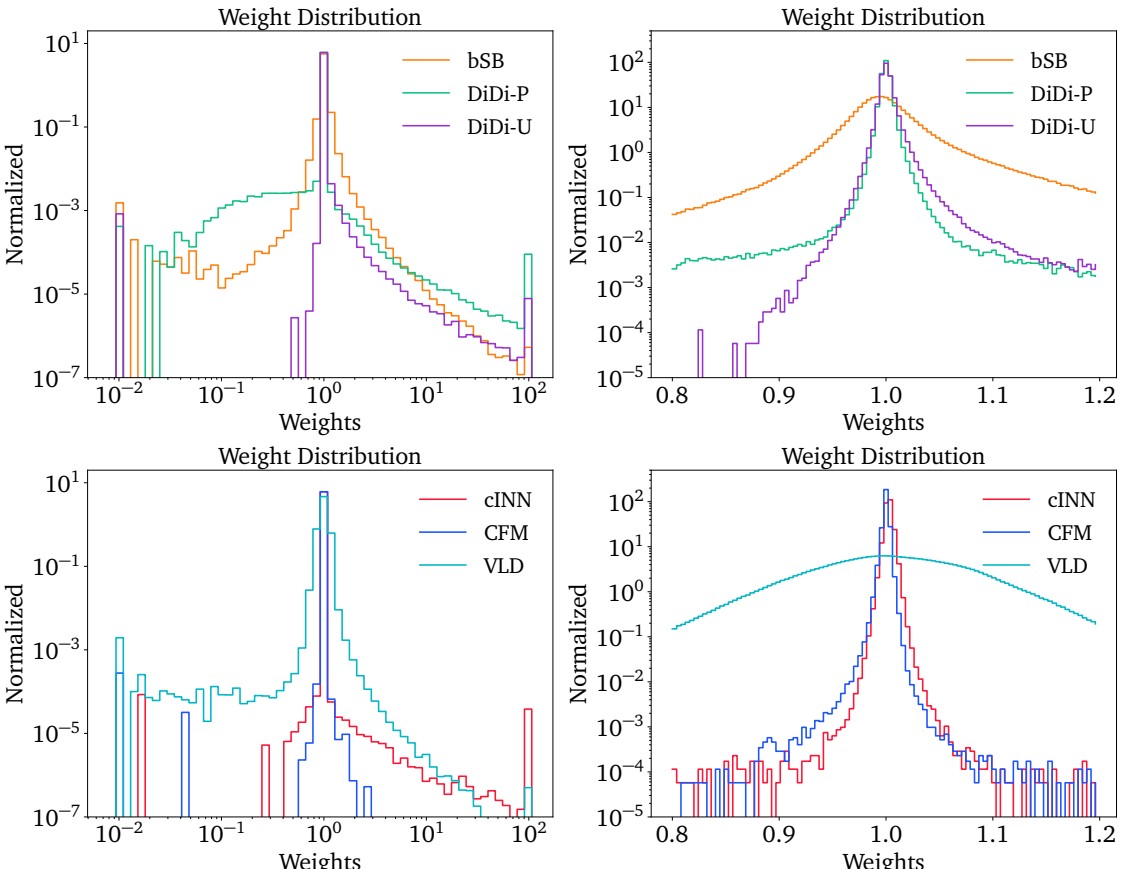

Figure 9: Classifier weight distribution on the Z+jets dataset. The top row shows the results for the distribution mapping methods, the bottom row for the generative methods. The left plots show the weight distribution over a large range, the right plots are zoomed in to resolve the area around $w = 1$.

from the hard scattering, before undergoing hadronization. While more physics assumptions/approximations are required for this type of unfolding, it is often performed by ATLAS and CMS [78–85]. Parton-level results are extremely useful, for instance, to combine measurements into a global analysis [2, 3], extract SM parameters [85, 86], or to compare new theory predictions without requiring these to be matched to parton showering programs [87].

The task is to map reco-level 4-momenta to parton-level 4-momenta defined by the $2 \to 2$ scattering and subsequent decays, in our case [34]

$$q\bar{q}/gg \to t\bar{t} \to (b\ell^+ \nu_\ell)(\bar{b}qq), \qquad \text{with} \quad \ell = e, \mu, \quad q = u, d, s, c, \tag{50}$$

plus the charge-conjugated process. The events are simulated with Madgraph5 3.4.2 [88] at $\sqrt{s} = 13$ TeV and with a top quark mass $m_t = 173$ GeV. One of the $W$ bosons decays leptonically, the other hadronically. Showering and hadronization are simulated with Pythia 8.306 [62], and detector effects with Delphes 3.5.0 [63] with the standard CMS card. We again reconstruct jets using the anti-$k_T$ algorithm [64], now with $R = 0.5$ and a $p_T$-dependent b-tagging efficiency. Leptons and jets are subject to the acceptance cuts $p_T > 25$ GeV and $|\eta| < 0.25$. We only keep events with exactly one lepton, at least 2 $b$-tagged jets, and at least two more jets.

This second benchmark process is technically more challenging than the $Z$+jets unfolding in terms of the six subjet observables, because of the higher phase space dimensionality and because we can no longer directly match reco-level and parton-level observables. To reconstruct the hard scattering, the network has to learn the non-trivial combinatorics as well as

complex correlations reflected in the intermediate mass peaks. We focus on a comparison of the different generative unfolding methods, which reproduce the forward simulation in their event-wise migration, but are most challenging from an ML-perspective. As before, we postpone the important question of model dependence to a later paper.

## 4.2 Generative unfolding

As a first attempt, we employ a straightforward phase space parametrization for the six top decay products,

$$
\begin{aligned}
&(p_{T,b_\ell}, \eta_{b_\ell}, \phi_{b_\ell}, \, p_{T,\ell}, \eta_\ell, \phi_\ell, \, p_{T,\nu}, \eta_\nu, \phi_\nu), \\
&(p_{T,b_h}, \eta_{b_h}, \phi_{b_h}, m_{q_1}, \, p_{T,q_1}, \eta_{q_1}, \phi_{q_1}, \, p_{T,q_2}, \eta_{q_2}, \phi_{q_2}).
\end{aligned} \tag{51}
$$

The lepton masses are common to all events, and we set them to zero at the level of our simulations. The bottom jets are generated with a common finite bottom mass. For the remaining jets, we have to keep track of the charm mass in the corresponding charm jets. This leads to a binary degree of freedom, in addition to the 18 standard phase space dimensions.

While at parton level all events have the same number of particles, at reco level we see a variable number of jets. Jets are produced in top and $W$-decays, but also in initial-state and final-state radiation, multi-parton interactions, underlying event, or pileup. Their number also strongly depends on the acceptance cuts. Naively, these additional jets are not expected to carry information on the hard process. However, they can sometimes cause events to pass selections by replacing top decay jets which do not pass the acceptance cuts, or lead to challenging event reconstruction due to jet combinatorics [58]. This means we cannot just ignore them.

While the standard cINN and CFM require a fixed-dimensionality condition, their transformer variants can handle variable dimensionality. Alternatively, we could employ an embedding network to overcome this limitation. Testing the impact of additional jets on our specific unfolding task with the Transfermer and TraCFM networks, we find that they do not benefit from additional reco-level jets significantly. Consequently, we restrict ourselves to a fixed maximum number of particles at reco-level for these networks. The particles we include in an ordered vector are the lepton, the missing transverse momentum, the two leading $b$-jets, and the two leading light-flavor jets. VLD does naturally includes all jets in an un-ordered fashion. The masses and transverse momenta of the particles are log-scaled before feeding them to the network.

We again use an RQS-cINN implementation from the FrEIA library [73], in the last block we replace the linear layers with Bayesian layers to track the network uncertainties. The CFM encodes the velocity in a standard MLP network. The time $t$ is embedded to a higher dimension using a random Fourier feature encoding [89] before being concatenated to the other network inputs, as we found that this improves results in higher-dimensional tasks. Following [90] we only make the last network layer Bayesian, as too many Bayesian weights can make the training of large networks unstable. For the two Transformer-based networks we employ the standard PyTorch Transformer implementation. The attention blocks are then followed by a single Bayesian RQS block or a single Bayesian linear layer for the Transfermer and the TraCFM respectively. For the TraCFM we employ the same time embedding as for the CFM and concatenate the encoded time to the transformer output before feeding it to the final layer.

The results for the cINN, its Transfermer variant, the CFM, and its TraCFM variant along with VLD are shown in Fig. 10. In the plots we focus on the challenging distributions, mainly the intermediate mass resonances and the angular correlations. We have checked that the rest of the kinematics is reproduced mostly at the percent level by the generative networks.

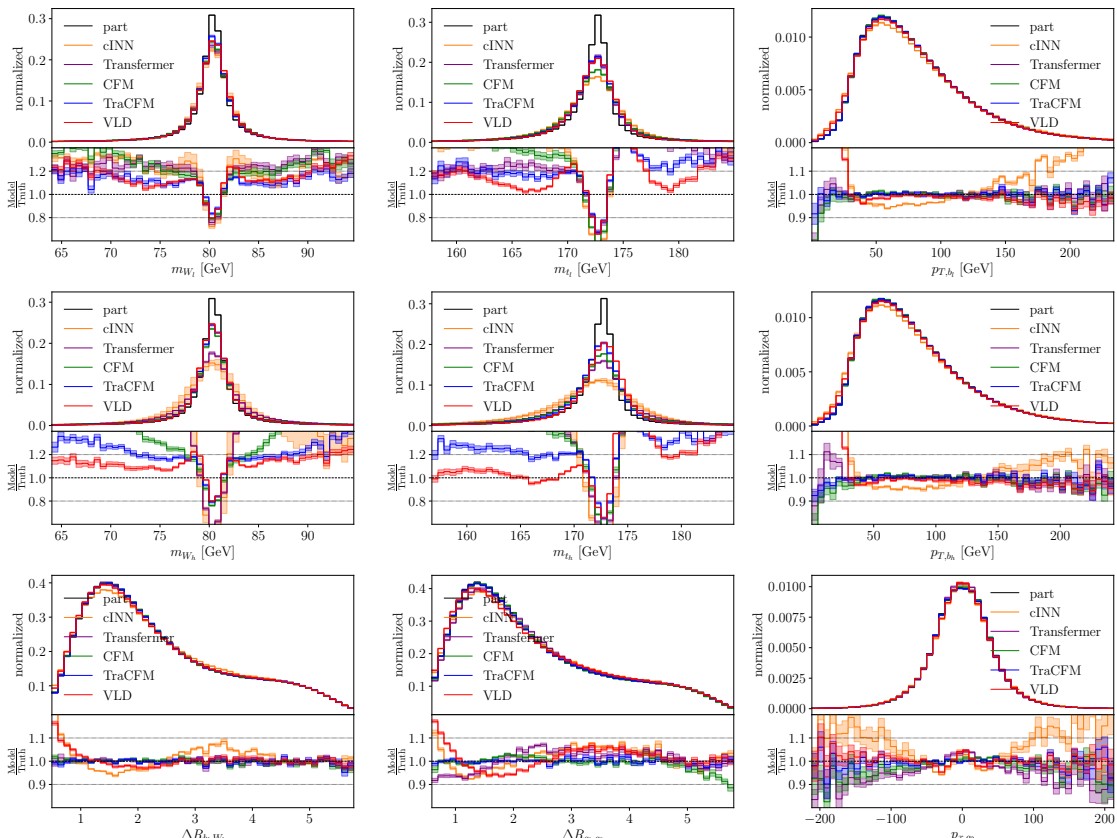

Figure 10: Unfolded top pair distributions from conditional generation using the naive phase space parametrization of Eq.(51). For the Bayesian cINN, Transfermer, CFM and TraCFM the error bars are based on drawing 50 samples. For the VLD the error bars are given by unfolding each event 128 times and showing the bin-wise mean and standard deviation.

Generally, we find that the lepton and neutrino kinematics are learned slightly better than the quark kinematics. As shown in the top rows, the correlations describing the intermediate particles are not learned as well. For the resonances, all networks struggle. Because they only require to correlate two independent 4-momenta, the $W$-peaks are learned a little better than the top peaks. Also, the leptonic decay is learned better than the hadronic decay. Altogether, the Transformer-enhanced networks perform better than the CFM, which in turn beats the cINN.

## 4.3 Generative unfolding using physics

The choice of phase space parametrization can be crucial for the performance of generative networks [91]. To solve the problems with intermediate on-shell propagators, we employ the dedicated top-mass parametrization proposed in Ref. [33]. It directly predicts the top and $W$-kinematics and makes the simpler decay kinematics accessible via correlations. As the phase space basis we choose the top 4-momentum in the lab frame, three components of the $W$ 4-momentum in the top rest frame, and two (three for the hadronic case) components of the first $W$-decay product in the $W$ rest frame,

$$
\begin{aligned}
&(m_t, p_{T,t}^L, \eta_t^L, \phi_t^L, m_W, \eta_W^T, \phi_W^T, \eta_\ell^W, \phi_\ell^W), \\
&(m_t, p_{T,t}^L, \eta_t^L, \phi_t^L, m_W, \eta_W^T, \phi_W^T, m_{q_1}, \eta_{q_1}^W, \phi_{q_1}^W).
\end{aligned} \tag{52}
$$

The superscripts $L, T, W$ denote the rest frames. We then employ a Breit-Wigner mapping using the mass values in the event generator

$$\sqrt{2} * \mathrm{erfinv}\left[\frac{2}{\pi}\arctan(m - m_{\mathrm{peak}})\right], \tag{53}$$

to turn the sharp mass peaks into a Gaussian-like shape.

The results with this paramerization are shown in Fig. 11. We drop the cINN and focus on the better CFM implementations. Now that the intermediate masses are directly predicted by the networks, we reproduce them within a few percent. The kinematics of the decay particles, now correlations between the directly predicted dimensions, are also faithfully modeled. Because the learning task has become easier, the difference between the CFM and the TraCFM is smaller. So physics helps, as it tends to.

Similar to Sec. 3.5, we again train a classifier to distinguish the generated events from the training data truth. In Fig. 12 we show the distribution of learned classifier weights for the three generative unfoldings. In this case, we see that while the one-dimensional kinematic distributions look similarly good for the three models in Fig. 11, there are significant differences in the precision with which the generative networks reproduce the multi-dimensional target distributions. The fact that all weight distributions are peaked around $w \approx 1$ and that the tails on the parton-level training and generated datasets are identical indicates that there is no definite failure mode [75]. On the other hand, the level of agreement is significantly improved,

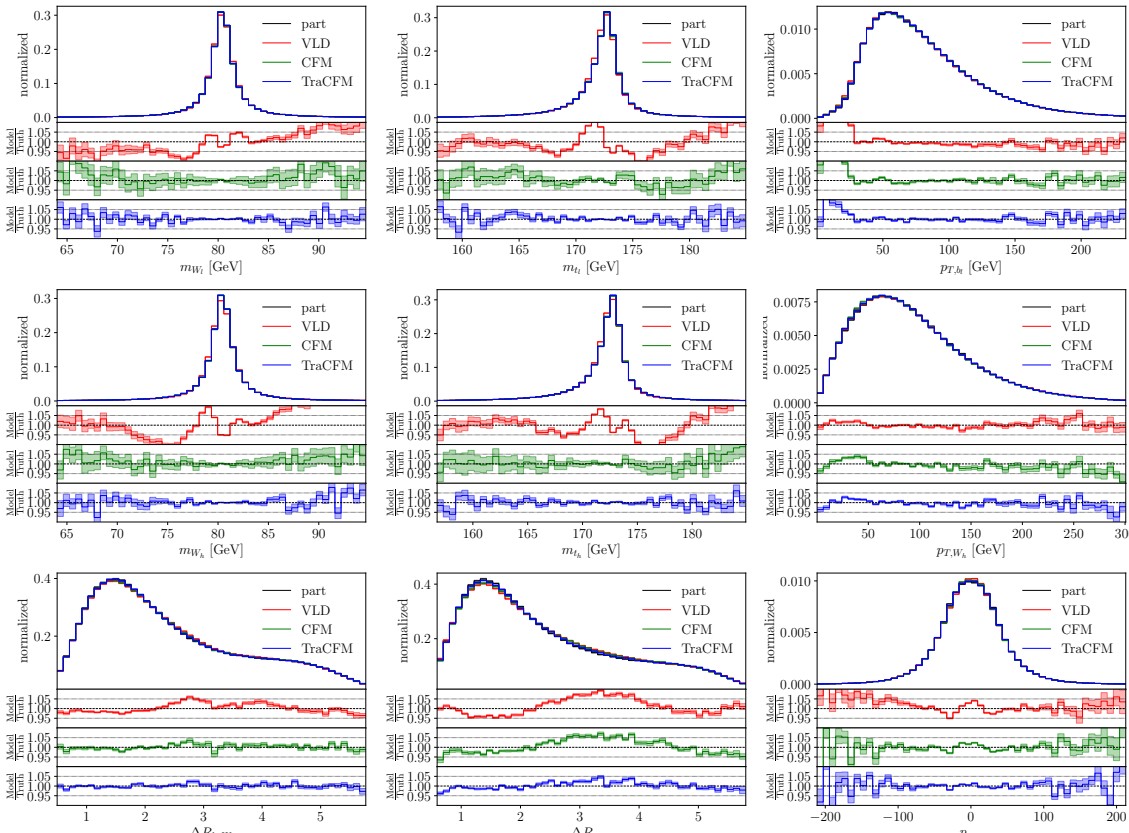

Figure 11: Unfolded top pair distributions from conditional generation using the dedicated phase space parametrization of Eq.(52). For the Bayesian cINN, Transfermer, CFM and TraCFM the error bars are based on drawing 50 amples. For the VLD the error bars are given by unfolding each event 32 times and showing the bin-wise mean and standard deviation.

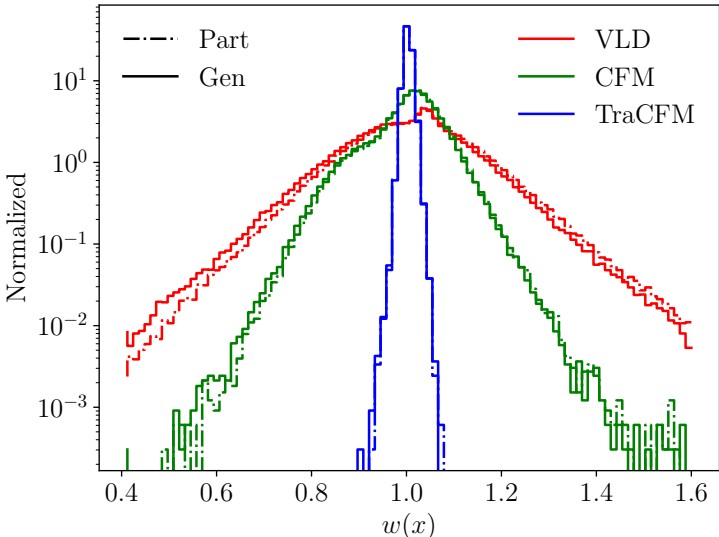

Figure 12: Weight distributions from a trained classifer between true and generated top pair events. The corresponding AUC values are 0.53 for the VLD, 0.51 for the CFM and 0.501 for the TraCFM.

going from the VLD to the CFM, and then adding the transformer feature of the TraCFM to encode combinatorics [58]. For the latter, we again reach the percent-level precision we observed for the $Z$+jets detector unfolding in Sec. 3.

# 5 Outlook

Machine learning is changing the face of LHC physics, and one of the most exciting developments is that it enables unbinned, high-dimensional, precise unfolding. This includes detector unfolding as well as inverting the first-principle simulations to the parton level. There exist three different ML-methods and tools for such an unfolding, (i) event reweighting or Omni-Fold, (ii) mapping distributions, and (iii) conditional generative unfolding. All these methods have been developed to a level, where they are ready to be further studied for use by the LHC experiments. In this paper, we give an overview of the different methods and corresponding tools, including an update to the most recent neural network architectures and a rough comparison of the strengths of the different methods.

Our first task is to unfold detector effects for a set of six subjet observables in $Z$+jets production. Here, reweighting-based unfolding, a supervised classification task, reproduces all true particle-level distributions and defines a precision benchmark shown in Fig. 3. A new Bayesian variant of OmniFold might provide complementary strengths to the existing method.

Alternatively, distribution mapping can be trained on matched events efficiently. We found that the (Bayesian) Schrödinger Bridge and Direct Diffusion implementations consistently provide high performance, shown in Fig. 6. Alternatively, distribution mapping can be trained on unmatched data, which limits its ability to reproduce the actual detector effects, but can be useful when one is missing matched training data.

Third, unfolding by learning and sampling conditional inverse probabilities is ideally suited to model complex detector effects, but also the most challenging network architecture. We have compared a series of tools, including invertible networks without and with a transformer encoding, as well as diffusion networks without and with a transformer, and with an enhanced

latent representation. In Fig. 7 we have shown that the conditional generative tools match the precision of distribution mapping. In addition, we have shown to which level the different methods learn the event migration or optimal transport defined by the forward detector simulation, rather than an abstract mapping defined by the network architecture.

Finally, we have applied our unfolding methods and tools to invert $t\bar{t}$ events to the hard process of top pair production with subsequent decays. Here, correlations pose a serious challenge, specifically the intermediate mass peaks. We have found that they can be learned precisely once we represent the phase space in a physics-inspired kinematic basis, as can be seen in Fig. 11. In addition to the physics pre-processing, the combination of a diffusion model with a transformer guaranteed the best performance among the conditional generative unfolding networks.

Altogether, we have shown a multitude of different methods and tools for ML- unfolding, with dedicated individual strengths.[2] All of them are ready to be studied further in the context of LHC analyses. Their complementarity is a strength for building confidence in advanced tools for high-dimensional cross section measurements. Future work will focus on how the different approaches handle prior dependence, backgrounds, and acceptance effects, as well as a comprehensive treatment of the uncertainties associated with these steps.

## Acknowledgements

We would like to thank Sofia Palacios Schweitzer for her crucial contributions to DiDi-unfolding and Bogdan Malaescu for many enlightening discussions on unfolding.

**Funding information**   NH, JMV and AB are funded by the BMBF Junior Group Generative Precision Networks for Particle Physics (DLR 01IS22079). TP and AB would like to thank the Baden-Württemberg-Stiftung for financing through the program *Internationale Spitzenforschung*, project *Uncertainties — Teaching AI its Limits* (BWST_IF2020-010). The Heidelberg group is supported by the KISS consortium (05D23GU4) funded by BMBF in the ErUM-Data action plan, the Deutsche Forschungsgemeinschaft (DFG, German Research Foundation) under grant 396021762 – TRR 257 *Particle Physics Phenomenology after the Higgs Discovery*, and through Germany's Excellence Strategy EXC 2181/1 – 390900948 (the *Heidelberg STRUCTURES Excellence Cluster*). SD, VM, and BN are supported by the U.S. Department of Energy (DOE), Office of Science under contract DE-AC02-05CH11231. DW, KG, and MF are supported by DOE grant DE-SC0009920. This research used resources of the National Energy Research Scientific Computing Center, a DOE Office of Science User Facility supported by the Office of Science of the U.S. Department of Energy under Contract No. DE-AC02-05CH11231 using NERSC award HEP-ERCAP0021099.

## A   Combined $Z$+jets results

In Fig. 13 we compare the unfolding results for $Z$+jets events, as discussed in Sec. 3. We show the same kinematic observables as in Fig 3 for the (b)OmniFold benchmark, in Fig. 6 for the distributions matching, and in Fig. 7 for the conditional generation. We omit all error bars representing statistical or Bayesian network uncertainties. The (b)OmniFold curves show learned densities from noisy data and cannot be directly compared to the other networks.

---

[2]Many of the codes used in this paper will be made publicly available, together with a set of tutorials accompanying Ref. [41].

None of the networks have been especially optimized for the task, so for all of them there should still be possible performance gains.



Figure 13: Results collected from Sec. 3, showing all unfolding networks, as well as the (b)Omnifold de-noising benchmark.

# B   Hyperparameters

Table 2: Network and training hyperparameters for the OmniFold and bOmniFold networks in Figs. 3, 4, and 5.

| Parameter | (b)OmniFold Py-to-Py | (b)OmniFold He-to-Py |
|---|---|---|
| Optimizer | Adam | |
| Learning rate | 0.001 | |
| LR schedule | Cosine annealing | |
| Batch size | 128 | |
| Epochs | 30 | 500 |
| Network | MLP | |
| Number of layers | 4 | |
| Hidden nodes | 80 | |
| Bayesian regularization | 1 | 1 |

Table 3: Network and training hyperparameters for the Direct Diffusion and CFM networks in Figs. 6, 7, 10, and 11

| Parameter | DiDi | CFM $Z$+jets | CFM $t\bar{t}$ | TraCFM |
|---|---|---|---|---|
| Optimizer | Adam | | | |
| Learning rate | 0.001 | | | |
| LR schedule | Cosine annealing | | | |
| Batch size | 16384 | | | |
| Epochs | 500 | 400 | 1000 | 500 |
| Network | MLP | MLP | MLP | Transformer |
| Number of layers | 8 | 8 | 8 | - |
| Hidden nodes | 512 | 512 | 1024 | - |
| Transformer blocks | - | - | - | 6 |
| Transformer heads | - | - | - | 4 |
| Embedding dim | - | - | - | 128 |
| Bayesian regularization | 1 | | | |

Table 4: Network and training hyperparameters for the cINN and Transfermer in Figs. 7 and 10.

| Parameter | cINN $Z$+jets | cINN $t\bar{t}$ | Transfermer |
|---|---|---|---|
| Optimizer | Adam | | |
| Max Learning rate | 0.0003 | | |
| LR schedule | One cycle | | |
| Batch size | 1024 | | |
| Epochs | 75 | 130 | 250 |
| Network | RQS-INN | RQS-INN | Transformer+RQS |
| INN blocks | 10 | 20 | 1 |
| RQS bins | 24 | 30 | 30 |
| Subnet layers | 5 | 5 | 5 |
| Subnet dim | 200 | 256 | 256 |
| Transformer blocks | - | - | 6 |
| Transformer heads | - | - | 4 |
| Embedding dim | - | - | 128 |

Table 5: Network and training hyperparameters for the Schrödinger Bridge in Fig. 6.

| Parameter | SB |
|---|---|
| Optimizer | Adam |
| Learning rate | 0.001 |
| Batch size | 128 |
| Network Updates | 250000 |
| Network | Fully connected ResNet |
| Blocks | 6 |
| MLP size | 256 |

Table 6: Network and training hyperparameters for the VLD networks in Figs 7, 10, and 11.

| Parameter | VLD $Z$+jets | VLD $t\bar{t}$ |
|---|---|---|
| Optimizer | Adam | Adam |
| Initial Learning rate | $5 \times 10^{-4}$ | $5 \times 10^{-4}$ |
| Fine-tune Learning rate | $1 \times 10^{-4}$ | $1 \times 10^{-4}$ |
| Batch size | 1024 | 1024 |
| Updates | 1 Million | 1 Million |
| Hidden Dimensions | 64 | 64 |
| Denoising Layers | 8 | 8 |
| Detector Encoder Layers | 6 | 6 |
| Part* Encoder Layers | 6 | 6 |
| Part* Decoder Layers | 6 | 6 |

Table 7: Network and training hyperparameters for the classifier networks in Figs. 9 and 12.

| Parameter | Z+jets | $t\bar{t}$ |
|---|---|---|
| Optimizer | Adam | |
| Learning rate | 0.001 | |
| LR schedule | Cosine annealing | |
| Batch size | 128 | |
| Epochs | 20 | 50 |
| Network | MLP | |
| Number of layers | 5 | |
| Hidden nodes | 256 | |
| Dropout | 0.1 | |

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
