# Peer review of "The Landscape of Unfolding with Machine Learning"

_SciPost Physics, doi:SciPost Phys. 18, 070 (2025)_

## Round 1 · Author Response

Answer to report 1

Dear Referee 1, Thank you very much for your positive feedback regarding our publication and thank you for the detailed suggestions. We carefully went through your list of suggested changes and implemented most of them.

  1. On page 3 in the introduction it read "access to the data and the detector simulation." it should read "access to the data and accurate detector simulation."

Answer: We changed this as suggested.

  1. Consider adding "Comparison of unfolding methods using RooFitUnfold, ArXiV:1910.14654" to the references where you reference [4-6]

Answer: We added this reference as suggested.

  1. Last line of page 3 reads "App. A just combine results from the Z+jets study in Sec. 3." where the word 'just' should be removed

Answer: We deleted the “just”.

  1. On page 4 under equation (1) P_gen is introduced. Replace with P_gen(x_part) to stick to the notation used in the rest of the document

Answer: We decided to leave this as is since p_gen(x_part) is already introduced with its argument in the paragraph above equation (1).

  1. Before equation (2) you’re missing a sentence to explain why the ration of the two x_reco’s isn’t enough and you need a classifier, e.g. if you want to go to an event-by-event space rather than say only something about the full collection of the data.

Answer: We are using the classifier to estimate the density ratio between the simulated and measured reco-level distributions. This method works statistically on the full collection of the data and has no event-by-event interpretation.

  1. In equation (4), what do the numbers between brackets mean in this diagram? it’s not clear and confusing since equations are number in those same brackets way, but you don't mean those.

Answer: The numbering was meant to indicate the order of the steps in the algorithm. We slightly changed the figure to make the order of the steps clear without numbering them. Thank you for pointing out that this was unclear.

  1. You have not introduced BNN yet as a acronym when used in the last line of page 4.

Answer: We wrote it out as suggested.

  1. In the first paragraph of section 2.2 it's written "based on the paired or unpaired simulated events" the pairing isn't explained yet so you need to add a sentence to explain that here

Answer: The pairing is explained in the first sentence below equation (1), we decided to not repeat it here.

  1. In section 2.2.1 and during the rest of the document the symbol ~ is used quite often. This is in mathematics used to indicate approximation, but here it seems to (sometimes) be used to indicate the mathematical symbol for ‘in set’. Please review this during the entire document and replace with correct symbol where done wrong.

Answer: We use that symbol for “distributed as”. This is a common usage in the literature.

  1. Below equation (7) you use sθ ( x , t ), which hasn't been defined yet.

Answer: We reformulated this sentence to make the definition clear.

  1. Equation (12) should be \mu_t(x0,x1) rather than just \mu_t

Answer: We changed this as suggested.

  1. In equation (18) and some more equations below there's a right arrow used. I don’t understand this, do you mean to say that this holds for decreasing values of t and the other for increasing values of t? Because then it basically collapses in eq (19) when you’re going down in values of t instantaneously. Do you maybe mean =? otherwise explain in text.

Answer: We use the arrow to indicate behaviour when approaching limits.

  1. Page 8 above equation (24) "droping " typo

Answer: We fixed this.

  1. I don’t understand the notation ((p(xpart|xreco))) below equation (27)

Answer: The double brackets were a mistake, we fixed this.

  1. After equation (42) you should remind the reader here what all the functions and parameters are.

Answer: We added a sentence and a reference to the definitions.

  1. Before section 3, this is where you add the explanation of the calculation and importance of the uncertainties on the unfolded distribution. I really think that before you start showing the comparisons in third and fourth section it would be useful to include a short section on the uncertainty estimation. In the rest of the document the authors use a certain number of redrawn or reweighed distributions, but never explain the number of samples you average over or why this would be a good error estimation (generally it is a decent one, but it still needs explaining). Additionally it needs a clarification that only statistical uncertainties are considered (if I understood that right) and that systematic effects such as from the detector modelling are ignored for this study but would increase the uncertainty and that the uncertainty is expected to cover unfolded/truth=1 etc.

Answer: The method we use for estimating uncertainties of most of the models are Bayesian neural networks. They give access to the training uncertainties of the neural network as well as the statistical uncertainties of the sampling. We covered this in the theory section. It is possible for these uncertainties to not cover the deviations from the truth if, for example, the implicit bias of the network is too restrictive to fit the density precisely everywhere. Our objective is to benchmark different ML-based unfolding techniques based on how precisely they recover the high-dimensional particle/parton level distributions they were trained on. We therefore only estimate the uncertainties associated with this step. Other sources of uncertainty, such as uncertainty on the detector model, are not included at this stage. For the VLD we are restricted to statistical uncertainties obtained by repeated sampling, as there does not exist a Bayesian version of this model yet. We made this point clearer in the text. As we write in the introduction, this is only the first step in a complete unfolding procedure. Future work will address the next steps, such as correcting for simulation-nature differences and including acceptance and efficiency effects. We will then also incorporate the uncertainties associated with these steps.

  1. Figure 3. is hard to read even with increasing the size on my screen. For those regions where you aren’t able to get an uncertainty estimate and your estimate of the central value is off, you should consider not showing the unfolded distribution at all or finding a way to estimate the uncertainty, because this way it implies a very good knowledge of the central values (e.g. small uncertainties) while in reality you know very little in these regions and aren’t doing well. you’re not staying inside your claimed percentage level either, and although you mention that in the text it might become misleading.

Answer: We think it is important to show the unfolded density over the entire range to allow the reader to judge the performance of the method. The alternative would mean to hide the areas where the methods do not perform as precisely as we want them too. We rephrased the text more conservatively.

  1. In the paragraph below equation (48) it's written " the agreement between the unfolded and true particle-level events is at the per-cent level or better. " that is not what i see in the plot… Consider writing this more conservatively.

Answer: We rephrased the text more conservatively.

  1. In the second to last paragraph of section 3.3.2, it's written "As before, the unfolded observables agree well between OmniFold and bOmniFold. " Do you show this anywhere? I don't see it.

Answer: We do not show these plots, as they are not crucial for the point we want to make. The interesting observation in this section is the structure of the weight distribution shown in Fig. 5, which is discussed in the last paragraph of the section. We reformulated the text to make this clearer.

  1. The same paragraph continues with "this model uncertainty is not intended to be covered by the Bayesian error estimate." Could you add a sentence to what users should do for an error estimate then?

Answer: We added a sentence to clarify this.

  1. In the paragraph under figure6 you write "precise to the per-cent level.", but now you have coverage, which I would consider way more important. So you can explicitly mention this useful fact.

Answer: This is mentioned in the next sentences.

  1. The last sentence of section 3.4 reads "VLD approach shows slightly larger deviations from the target distributions." However, more importantly, VLD doesn’t seem to have coverage… Can you comment on that? why would that happen while the others are fine? what in the architecture would cause that? and what should a used do to avoid this problem or should it just not be used?

Answer: As mentioned in the reply to 16., the VLD, unlike the other models, is not implemented as a Bayesian network and therefore does not allow access to the network training uncertainties. We only show statistical uncertainties obtained from repeated sampling of the posterior distribution. This explains the smaller error bands for the VLD as compared to the other networks. We added a sentence in the paragraph before to clarify this.
Aside from the uncertainties, the deviation from the truth is slightly larger for the VLD, which could indicate that the other architectures are slightly more suited for this specific task.

  1. In section 4.2 there several times the typo "Transfermer"

Answer: The architecture is actually called “Transfermer”. We followed the name given to it by the authors of the cited reference. (https://arxiv.org/abs/2310.07752)

  1. In section 4.2 in the description of figure 9 it's written that "We have checked that all generative networks reproduce the kinematics of the top decay products at the percent level. " However, this doesn’t seem to be true, there’s more than 20% differences in figure 9. what claim do you mean to make here?

Answer: You are correct in pointing out that this sentence is misleading. It was meant to say that we show the difficult correlations in the plots, and that the “easy” kinematics of the decay particles, which are not shown, are learned to percent level precision. We rewrote this paragraph to make this clear.

  1. In the final paragraph of section 4.3 it's written "while the one-dimensional kinematic distributions look similarly good in Fig. 10 " i’m not sure i agree here, the coverage of your uncertainties is poor and clearly the distributions are more closely matched than in fig 9, although you also see clearly that structures in the unfolded distribution are inserted by the network.

Answer: This “similarly good” compares the three models in Fig. 10, not the results from Fig. 10 to those from Fig. 9. We reformulated to make this clearer.

  1. In the second to last paragraph of the outlook it's written " We have found that they can be learned precisely once we represent the phase space in a physics-inspired kinematic basis, as can be seen in Fig. 10. " this might be too strong of a claim considering fig10 and comments above

Answer: This statement explicitly refers to the final results in the physics-inspired phase space parametrization. We think it is a fair claim and is supported by the results shown in Fig 11 (was Fig 10 in v1). Especially the CFM-based models learn the density to high precision over the entire phase space.

  1. In the outlook, also mention the dependence on the training data (bias) that still needs to be studied in a future paper here. You have mentioned it before of course, but this definitely part of the outlook.

Answer: We added a sentence on this in the last paragraph of the outlook.

Thank you once again for taking the time to give us so detailed feedback Sincerely, the authors

Answer to report 2

Dear Referee 2, Thank you very much for your positive feedback regarding our publication. We carefully went through your list of suggested changes and addressed them.

  1. In Section 2.1, when introducing OmniFold, the description of pulling/pushing weights and the connected Eq.(4) are a bit sloppy. I understand the authors' intention to not fully reiterate the OmniFold paper and keep the introduction short. It might be helpful to use the introduced numbers in Eq.(4) to make better sense, as these numbers have been introduced but not used yet.

Answer: We changed Eq. (4) to make it clearer.

  1. In Section 2.2., where the authors mention previous references attempting to do these direct distribution mappings, they should also mention their own work on this, i.e. [1912.00477, 2006.06685].

Answer: In Section 2.2 we describe methods that aim at morphing one distribution into another, without respecting any pairing information between individual events. The works you are citing here are more closely related to the conditional generative approach discussed in section 2.3 and are cited in this context.

  1. In Section 2.3, in Eq.(27), having a direct arrow from sim to gen is confusing, as it was also done in the direct mapping before. This wrongly leads to the assumption that the generative models do the same thing. Maybe this can be done more clearly by introducing the "auxiliary" latent space z from which the mapping onto pgen and punfold is done, conditioned on the reco level.

NH: We understand the concern, but in the interest of making these diagrams compact we decided to keep it this way.

  1. On the same line, I would replace Eq.(29) with some Eq.(31)-equivalent, making clear what the input of the mapping G is. Then, for the cINN, you can either keep Eq.(31) and stress that this mapping is now invertible or mention this in the text while referring to the updated version of Eq.(29).

Answer: We changed this as suggested.

  1. I would avoid calling the transformer-enhanced cINN a "Transfermer" here. This naming was introduced in your MEM paper, where the architecture parametrized the forward direction, i.e. the transfer function. Here, however, you parametrize the inverse direction, i.e. the unfolding function. So you might want to call it "Transfolder" to stay consistent. However, to be more consistent with the TraCFM in the next section, I would call it "Tra-cINN" and consistently replace it everywhere in the text and the plots.

Answer: We understand the concern, but decided to stick with the name given to the architecture by the authors of the cited reference.

  1. Speaking about the TraCFM: Why is this architecture the only one that got a figure for illustration? I would either add one for the Transformer-cINN or skip both and refer both to the MEM paper.

Answer: This decision was made based on the results. The TraCFM architecture was found to be the strongest model and we therefore invested more time in explaining it.

  1. In Section 3.2, you compare OmniFold with bOmniFold in Table 1. Given the other methods shown later, I would like to see the same table, including all other methods introduced, for a better overview and benchmark comparison. On the same line, I would move Figure 12 in the appendix into the main body. This figure really is what somebody interested in comparing all the methods shown wants to see.

Answer: This figure is slightly misleading as it implies that the Omnifold results can be directly compared to the other models. As they were not solving the exact same task (Omnifold was working on a noisy version of the dataset), this is not the case. For this reason we do not show the direct comparison in the main body of the paper. We made this more clear in the text.

  1. In Section 3.5, you discuss event mitigation for different observables. Given the complexity of the data, it would also be interesting to show calibration curves as done in the first cINN paper [2006.06685]. Further, you mentioned that you had done the all-beloved classifier test to further assess the generative models' performance. Similar to the later shown Figure 11, I would like to see the trained classifier's weight distributions to better understand how the different generative methods in Section 3 compare.

Answer: Here, the focus was to showcase the difference between the generative models and the distribution mapping models. As the distribution mapping models do not learn a proper posterior distributions, calibration curves are not applicable there. For generative models they have been conclusively studied in the work that you cite. Concerning the classifier plot for Section 3, we followed your suggestion and added it to the paper.

  1. In Section 4.3, you mention that better preprocessing helps increase the sensitivity on critical phase-space observables like masses and cite a proposed parametrization. In this context, I would also cite the ELSA paper of one of your authors [2305.07696], which illustrates how different parametrizations affect the performance of generative models.

Answer: We followed your suggestion and added this reference.

  1. You show this nice Figure 11 in Section 4.3. To make it clear again, I would like to see that plot for Section 3 and all the generative methods shown there.

Answer: We followed your suggestion and added it to the paper in section 3.6.

  1. In your outlook, you mention that you only made a "rough" comparison of the methods. I believe, if you put Figure 12 into the main body, making Table 1 for all methods, adding calibration curves in Section 3, as well as adding a Figure 11-like plot for Section 3, your comparison will be much more detailed and helpful.

Answer: We addressed your suggestions concerning Fig. 12 and calibration curves in points 7 and 8. Concerning the classifier plot for Section 3, we followed your suggestion and added it to the paper.

  1. In Appendix B, for completeness, I would also add all hyperparameters used for (b)OmniFold.

Answer: Thank you for pointing this out. We added the table.

Thank you once again for taking the time to review our publication. Sincerely, the authors

Answer to report 3

Dear Referee 3, Thank you very much for your positive feedback regarding our publication. We carefully went through your list of suggested changes and addressed them.

1 - I find the title of section 2.1 with '(b)Omnifold' confusing - it reads like a typo, perhaps better to say 'Omnifold and bOmnifold'

Answer: We changed this as suggested.

2 - The notation with (1) through (4) in Eq 4 is not explained and confusing.

Answer: The numbering was meant to indicate the order of the steps in the algorithm. We slightly changed the figure to make the order of the steps clear without numbering them. Thank you for pointing out that this was unclear.

3 - I would suggest a dedicated (sub)section in Section 2 on source of uncertainty and biases in unfolding and their evaluation strategy. You already have a sentence somewhere that e.g. biases due to modelling distributions are not covered. This could be expanded into a section with sources of uncertainty (sample statistics, modelling sample statistics, various method uncertainties), and explain which of these are included in the shown comparison, and how they are evaluated, and which are not (including those that you say you will cover in a future paper)

Answer: We agree that a comprehensive treatment of uncertainties is essential for any unfolding algorithm. However this is beyond the scope of this paper. Our objective is to benchmark different ML-based unfolding techniques based on how precisely they recover the high-dimensional particle/parton level distributions they were trained on. We therefore only estimate the uncertainties associated with the network training. As we write in the paper, this is only the first step in a complete unfolding procedure. Future work will address the next steps, such as correcting for simulation-nature differences and including acceptance and efficiency effects. We will then also incorporate the uncertainties associated with these steps.

4 - On this topic, it is also good comment to what extent the uncertainty evaluation methods have good coverage. In the traditional unfolding methods, good coverage is not an automatic feature of many methods, in particular when regularisation is applied.

Answer: As you correctly point out, good coverage is not an automatic feature of unfolding algorithms, this applies to ML-based algorithms as well. The Bayesian neural networks, as mentioned in the reply to 3., estimate the uncertainties stemming from the network training and the limited training statistics. It is possible, and observed in some of our results, that these uncertainties do not cover the deviations from the truth if, for example, the implicit bias of the network is too restrictive to fit the density precisely everywhere.

5 - You can be more consistent in the metrics shown for the various methods, e.g. you show weight distributions in Section 3.2 and 4.3 but not elsewhere. A consistent set of plots would allow for a better comparison by the reader.

Answer: We agree with this point and added the weight distribution for the first dataset in Section 3.6. We do not show it in Section 4.2, as this is an intermediate step leading up to the final results on this dataset in Section 4.3.

6 - Figure 7 shows rather significant biases for the VLD method, relative to the shown uncertainty. I think this merits some more discussion than is currently in the paper (see also my point here on the consistency/comparability of shown uncertainties)

Answer: The VLD, unlike the other models, is not implemented as a Bayesian network and we therefore have no access to the network training uncertainties. The shown uncertainties for the VLD are only statistical uncertainties obtained from repeated sampling. We made this point clearer in the discussion of the results. It is possible for these uncertainties to not cover the deviations from the truth if, for example, the implicit bias of the network is too restrictive to fit the density precisely everywhere.

7 - Figure 9 shows distributions with very significant deviations from unity, yet the sentence in the body text says "reproduce at the percent level". This statement is not trivially apparent from Fig 9, given the strong deviation (w.r.t the shown uncertainty) of many of the methods in several of the plots. Can you explain better how you have come to this conclusion.

Answer: You are correct in pointing out that this sentence is misleading. It was meant to say that we show the difficult correlations in the plots, and that the “easy” kinematics of the decay particles, which are not shown, are learned to percent level precision. We rewrote the paragraph to make this clear.

Thank you once again for taking the time to review our publication. Sincerely, the authors

---

## Round 1 · List of Changes

See "Author comments".
We made no further changes that are not listed as responses to reviewer comments.

---

## Editorial Decision

published